# Dominant spinal muscular atrophy linked mutations in the cargo binding domain of BICD2 result in altered interactomes and dynein hyperactivity

Hannah Neiswender, Jessica E Pride, Rajalakshmi Veeranan-Karmegam, Phylicia Allen, Grace Neiswender, Avneesh Prabakar, Caili Hao, Xingjun Fan, Graydon B Gonsalvez*

Cellular Biology and Anatomy, Medical College of Georgia, Augusta University, Augusta, United States

## eLife Assessment

In their study, Neiswender et al. provide **important** insights into how BicD2 variants linked to spinal muscular atrophy alter dynein activity and cargo specificity. The authors present **convincing** evidence that disease-associated mutations lead to interactome changes, supported by additional validation of the BicD2/HOPS complex and discussion of their functional implications. This well-executed study offers invaluable datasets and a strong foundation for future exploration of disease mechanisms.

*For correspondence: ggonsalvez@augusta.edu

Competing interest: The authors declare that no competing interests exist.

**Abstract** Cytoplasmic dynein-1 (dynein) is responsible for the transport of most cellular cargo towards the minus end of microtubules. Dynein activation requires the multi-subunit dynactin complex and an activating cargo adaptor. The adaptors serve to link dynein with cargo and to fully activate the motor. Mutations in one of these activating adaptors, Bicaudal-D2 (BICD2), are associated with a neurodegenerative disease called Spinal Muscular Atrophy with Lower Extremity Predominance (SMALED2). The molecular defect that underlies SMALED2 is largely unknown. In addition to interacting with dynein, BICD2 has also been shown to associate with KIF5B, a plus-end directed microtubule motor. We hypothesized that interactome changes associated with mutant versions of BICD2, and the resulting differences in cargo transport, might underlie the etiology of SMALED2. To test our hypothesis, we first defined the interactome of wild-type human BICD2. This led to the identification of known BICD2 interacting proteins in addition to potentially novel cargo such as components of the HOPS complex, a six-subunit complex involved in endo-lysosomal trafficking. We next determined the interactome of three SMALED2-linked mutants in BICD2, two of which reside in the cargo binding domain. Interestingly, all three mutations resulted in BICD2-mediated dynein hyper-activation. Furthermore, all three mutants were associated with interactome changes. One of these mutants, BICD2_R747C, was deficient in binding to HOPS complex components and the nucleoporin RANBP2. In addition, this mutant also resulted in a gain-of-function interaction with GRAMD1A, a protein localized to the endoplasmic reticulum. This gain-of-function interaction resulted in mislocalization of GRAMD1A in BICD2_R747C expressing cells. Collectively, our results suggest that dynein hyperactivity, interactome changes, and cargo transport defects might contribute to the symptoms associated with SMALED2.

## Introduction

The intracellular transport of mRNAs, proteins, vesicles, and organelles is facilitated by microtubule motors. These motors operate along microtubules, which are polarized cytoskeletal filaments. Typically, kinesin family motors carry cargo toward the microtubule plus end (*Yildiz, 2025*), whereas transport toward the minus end relies primarily on a single motor protein, cytoplasmic dynein-1 (dynein; *Canty and Yildiz, 2020*; *Reck-Peterson et al., 2018*). While motor-driven transport is vital across various cell types, it holds particular importance in large, polarized cells like neurons. For instance, the axon of some neurons can extend over a meter in length, making them highly reliant on efficient transport systems. Even minor disruptions in motor-based transport can impair neuronal function and lead to disease (*Franker and Hoogenraad, 2013*). As a result, mutations affecting motor proteins, their activators, or adaptors are linked to a variety of neurological disorders (*Sleigh et al., 2019*).

The isolated dynein motor is present in an inhibited conformation and has a limited capacity to traverse the microtubule cytoskeleton (*Torisawa et al., 2014*; *Zhang et al., 2017*). Processive movement by dynein requires the multi-subunit dynactin complex and an activating cargo adaptor (*McKenney et al., 2014*; *Schlager et al., 2014*; *Splinter et al., 2012*). Bicaudal-D (BicD), initially identified in *Drosophila*, is one of the best characterized activating cargo adaptors (*Hoogenraad and Akhmanova, 2016*; *Mohler and Wieschaus, 1986*). Mammals encode four orthologous proteins, BICD1, BICD2, BICDR1, and BICDR2. The closest mammalian ortholog of *Drosophila* BicD is BICD2. BICD2 contains three coiled-coil regions designated CC1, CC2, and CC3 (*Figure 1A*; *Olenick and Holzbaur, 2019*). The first coiled-coil, CC1, is responsible for binding to dynein and dynactin (*Chowdhury et al., 2015*; *Urnavicius et al., 2015*), whereas the third coiled-coil, CC3, mediates cargo binding (*Hoogenraad and Akhmanova, 2016*). In the absence of cargo, an intramolecular interaction between the CC1 and CC3 regions of BICD2 results in the protein folding into an inhibited conformation (*Liu et al., 2013*; *Terawaki et al., 2015*; *Wharton and Struhl, 1989*). In this state, BICD2 is unable to bind dynein (*Figure 1B*). Cargo binding to CC3 competes with the intramolecular interaction, effectively opening up BICD2 and enabling the cargo-bound adaptor to interact with and activate dynein for motility (*Goldman et al., 2019*; *Huynh and Vale, 2017*; *Liu et al., 2013*; *McClintock et al., 2018*; *Sladewski et al., 2018*). This type of regulation ensures that the motility of dynein is tightly coupled to cargo binding.

The importance of BICD2 in dynein-mediated transport is highlighted by the fact that mutations in this gene are associated with a type of spinal muscular atrophy (SMA). SMA refers to a group of disorders characterized by progressive muscle weakness and atrophy caused by loss of motor neurons. Most often, SMA is autosomal recessive, and in the majority of these cases, causative mutations are linked to the SMN (survival of motor neurons) gene (*Angilletta et al., 2023*). Autosomal dominant SMA, caused predominantly by mutations in BICD2, is associated with weakness and atrophy of muscles in the feet and legs (*Koboldt et al., 2020*; *Neveling et al., 2013*; *Oates et al., 2013*; *Peeters et al., 2013*). Thus, this type of SMA is referred to as Spinal Muscular Atrophy with Lower Extremity Predominance (SMALED2). In some, the disease is relatively mild, whereas in others, it is more severe, presenting with contractures, hip dysplasia, brain abnormalities, cognitive impairment, and even death (*Koboldt et al., 2020*).

Although the link between BICD2 and SMALED2 has been recognized, we have a limited understanding of the molecular basis of the disease. Three SMALED2-associated mutations within the dynein binding region of BICD2 have been characterized. These mutations have been shown to increase the interaction between BICD2 and dynein/dynactin (*Huynh and Vale, 2017*). Consequently, dynein displayed hyperactivity in motility assays (*Huynh and Vale, 2017*). However, numerous additional SMALED2 mutations have also been identified within the cargo binding domain of BICD2 (*Koboldt et al., 2020*; *Martinez-Carrera and Wirth, 2015*). The mechanism by which these mutations result in SMALED2 is unknown. Given that BICD2 is a cargo adaptor for dynein, we hypothesized that mutations within the cargo binding domain likely result in interactome changes. As such, cargo transport defects caused by altered BICD2 interactomes might underlie the etiology of SMALED2. The goal of this study was to define the interactome of wild-type and mutant alleles of BICD2 and to identify potential interactome changes.

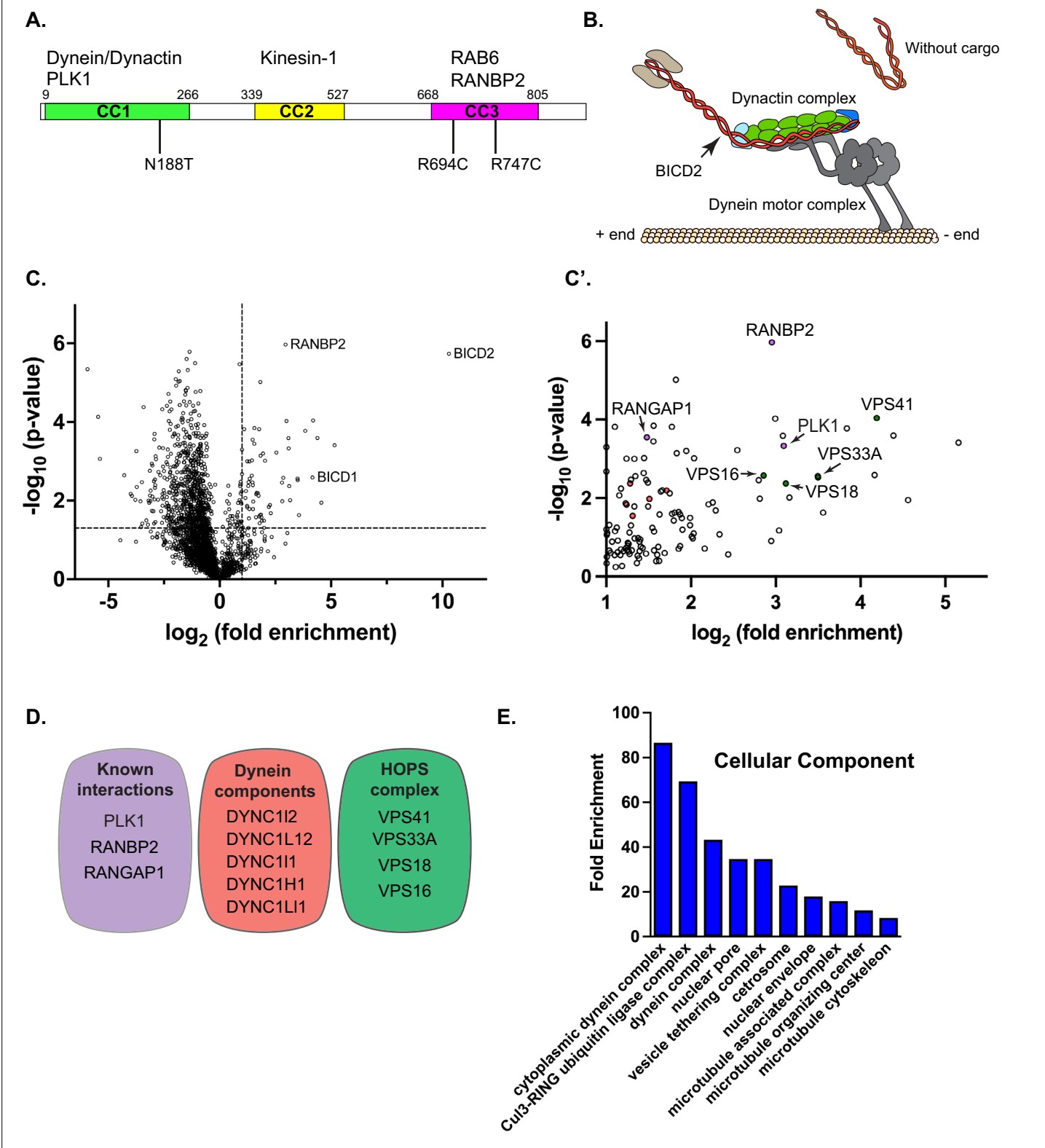

**Figure 1.** Wild-type BICD2 interactome. (**A**) Schematic of human BICD2, indicating the three coiled-coil domains and sites of interaction with dynein/dynactin, kinesin-1, and cargo. (**B**) Model of BICD2 binding to dynein/dynactin in a cargo-dependent manner. (**C, C'**) A volcano plot indicating proteins that were enriched with BICD2-mTrbo in comparison to RFP-mTrbo. The dashed line along the x-axis indicates a fold enrichment of 2, whereas the dashed line along the y-axis indicates a p value of 0.05. Specific interacting proteins were considered those that were enriched at least twofold with BICD2-mTrbo and had a p value of at least 0.05. These candidates are shown in the zoomed-in image in C'. (**D**) Known BICD2 interacting proteins,

*Figure 1 continued on next page*

*Figure 1 continued*

components of the dynein motor, and components of the HOPS complex that were specifically enriched with BICD2-mTrbo are indicated. (**E**) A cellular component GO analysis of the BICD2 interactome.

The online version of this article includes the following source data and figure supplement(s) for figure 1:

**Figure supplement 1.** HEK cells expressing the indicated constructs were transfected with a plasmid expressing GTP-locked RAB6A (GFP-RAB6A Q72L).

**Figure supplement 1—source data 1.** Original files for western blots displayed in *Figure 1—figure supplement 1*.

**Figure supplement 1—source data 2.** PDF file for western blots displayed in *Figure 1—figure supplement 1* with the bands marked.

## Results

### Defining the wild-type BICD2 interactome

In order to test our hypothesis that SMALED2-associated mutations in the cargo binding domain of BICD2 are associated with interactome changes, it was first critical for us to define the interactome of wild-type BICD2. Although BICD2 has been known to function as a dynein cargo adaptor, only a few proteins have been shown to interact with BICD2 (*Hoogenraad and Akhmanova, 2016*; *Olenick and Holzbaur, 2019*). A more comprehensive interactome of BICD2 was defined by Redwine and colleagues using proximity biotin ligation (*Redwine et al., 2017*). However, the goal of that study was to more broadly define the dynein interactome, and potential novel BICD2 interacting proteins were not validated by follow-up studies.

To begin our analysis, we generated stable cell lines expressing either RFP-tagged mini-TurboID (RFP-mTrbo) or full-length BICD2 tagged on the C-terminus with mini-TurboID (BICD2-mTrbo). TurboID is a proximity biotin ligase that was generated by selecting mutants of BioID that displayed much higher levels of activity in comparison to the original enzyme (*Branon et al., 2018*). mini-TurboID is a smaller version of TurboID, the biotinylation activity of which is more tightly coupled to the addition of exogenous biotin (*Branon et al., 2018*). HEK293 FLP-In T-Rex cells were used for this experiment because this enabled us to integrate the constructs at the same genomic locus across all cell lines. In addition, expression of the fusion proteins was inducible upon the addition of tetra-cycline. This prevents any toxic effects that might arise from constitutive expression of SMALED2 alleles of BICD2.

Cells expressing either RFP-mTrbo or BICD2_WT-mTrbo were grown to scale and proximal proteins were labeled using biotin. Biotinylated proteins were purified using streptavidin magnetic beads and identified using mass spectrometry, with the entire experiment being conducted in triplicate. Proteins enriched in the BICD2_WT-mTrbo pellet at least twofold in comparison to the RFP-mTrbo pellet and having a p value of at least 0.05 were considered potential BICD2 interacting partners (*Figure 1C and C'*; *Supplementary file 1*). This list, which consists of 66 proteins, included several dynein components and known BICD2 interacting proteins such as RANBP2, RANGAP1, and PLK1 (*Figure 1D*; *Gallisà-Suñé et al., 2023*; *Splinter et al., 2010*). In addition, four components of the HOPS complex were also specifically enriched in the BICD2_WT-mTrbo pellet (*Figure 1D*). The HOPS complex consists of six subunits and is involved in the fusion of late endosomes with lysosomes (*van der Beek et al., 2019*). Consistent with known BICD2 functions, a cellular component GO analysis indicated an enrich-ment of terms such as 'cytoplasmic dynein complex', 'nuclear pore', and 'vesicle tethering complex' (*Figure 1E*).

Surprisingly, we failed to identify RAB6A in our interactome analysis. RAB6A was the first cargo identified for BICD2 using a yeast two-hybrid screen (*Matanis et al., 2002*). Previous studies have shown that GTP-bound RAB6A is the preferred binding partner for BICD2 (*Matanis et al., 2002*). Thus, in order to determine whether RAB6A is capable of associating with full-length BICD2-mTrbo, we transfected cells expressing either RFP-mTrbo, full-length BICD2-mTrbo, or a version of BICD2-mTrbo lacking the CC3 domain along with a GTP-locked mutant of RAB6A (GFP-RAB6A Q72L). Consistent with published results, GFP-RAB6A Q72L specifically associated with full-length BICD2-mTrbo (*Figure 1—figure supplement 1*). Based on this, we conclude that although a GTP-locked version of RAB6A is capable of interacting with BICD2-mTrbo, the interaction between endogenous RAB6A and BICD2 might not be that prevalent under physiological conditions.

## Subunits of the HOPS complex associate with BICD2 in vivo

Four components of the six-member HOPS complex were found to associate with BICD2 in our interactome analysis, with VPS41 being the fifth most enriched protein (*Figure 1C' and D*; *Supplementary file 1*). We therefore chose to validate these interactions using co-immunoprecipitation. Strains expressing either RFP-mTrbo, full-length BICD2-mTrbo, or the delta CC3 domain construct (BICD2_delCC3-mTrbo) were used for this analysis. The TurboID constructs contain a V5 tag. Thus, V5 trap beads were used to immunoprecipitate the tagged proteins. Bound proteins were eluted and analyzed by western blotting using antibodies against VPS41 (*Figure 2A*), VPS18 (*Figure 2B*) or VPS16 (*Figure 2B*). Consistent with our interactome analysis, all three HOPS components specifically co-immunoprecipitated with full-length BICD2-mTrbo (*Figure 2A and B*). A similar interaction between BICD2-mTrbo and VPS41 was also detected in cells depleted of endogenous BICD2 (*Figure 2—figure supplement 1A*). Thus, we conclude that BICD2 associates with VPS41, VPS18, and VPS16 in vivo and that these interactions require the BICD2 cargo binding domain.

Previously identified BICD2 cargo such as RANBP2 and RAB6A have been shown to interact with the isolated CC3 domain of BICD2 (*Matanis et al., 2002*; *Splinter et al., 2010*). In order to determine whether the same conditions apply for the HOPS complex components, we repeated the experiment using the same three strains mentioned above in addition to a strain expressing the isolated CC3 domain fused to mTrbo (BICD2_CC3-mTrbo; *Figure 2C*). Unexpectedly, although RANBP2 was able to interact with both full-length BICD2 and the isolated CC3 domain, VPS41, VPS18, and VPS16 were only capable of associating with full-length BICD2 (*Figure 2—figure supplement 1B*). Thus, the minimal cargo binding domain of BICD2 is not sufficient for interacting with the HOPS complex components. We also attempted the binding experiment using an extended BICD2 C-terminal construct (residues 442–824), but even this construct failed to efficiently bind the HOPS complex components (data not shown). Finally, we generated a BICD2 construct that lacked the first coiled coil domain (BICD2_delCC1, residues 288–824). Although this construct was able to bind VPS41 above background levels, the binding was reduced in comparison to the full-length protein (*Figure 2—figure supplement 1*). RANBP2 binding was also reduced with this deletion construct (*Figure 2—figure supplement 1C*). It is possible that this deletion affects the normal folding of BICD2 or that it results in the protein folding into a locked conformation that inhibits cargo binding. Collectively, our results suggest that although the CC3 domain of BICD2 is required for interaction with the HOPS complex, it is not sufficient.

Among the HOPS components, VPS41 was by far the most enriched in the BICD2 interactome (*Figure 1C and C'*). We therefore hypothesized that the interaction between BICD2 and the HOPS complex is mediated by VPS41. Contrary to our hypothesis, however, siRNA-mediated depletion of VPS41 had little effect on the ability of BICD2 to interact with VPS16 and VPS18 (*Figure 2—figure supplement 1D*). This suggests that BICD2 is able to independently associate with multiple HOPS components, potentially interacting with a common motif or domain present within these proteins. Our inability to map the minimal domain of BICD2 required for interaction with the HOPS complex components precludes our ability to test for their direct interaction. Thus, although our results indicate that BICD2 associates with these proteins in vivo, we cannot conclude whether the interaction is direct or indirect.

## The proper localization of VPS41 and LAMP1 vesicles requires BICD2

We next examined the localization of the HOPS complex in control versus BICD2-depleted HeLa cells. Although antibodies are available that can detect endogenous VPS41, VPS18, and VPS16 by western blot, these antibodies were not suitable for detecting the native protein by immunofluorescence. Thus, we used a GFP-tagged VPS41 construct for this analysis. In HeLa cells, microtubule minus ends are present at the perinuclear centrosome and plus ends extend towards the cortex. As expected, given the role of the HOPS complex in late endosome-lysosome fusion, GFP-VPS41 positive vesicles were enriched in a perinuclear area. Dispersed vesicles and vesicles present closer to the cortex were also observed in cells transfected with a control siRNA (*Figure 3A, D and E*). As expected, depletion of Dynein heavy chain (DHC, DYNC1H1) resulted in a reduction of perinuclear clustering and an accumulation of vesicles close to the cell cortex (*Figure 3B, D, E*, *Figure 3—figure supplement 1A*). Because BICD2 is a cargo adaptor for dynein, we expected that depletion of BICD2 would produce a similar phenotype. Surprisingly, however, BICD2 depletion resulted in perinuclear clustering of GFP-VPS41 vesicles (*Figure 3C, D, E* and *Figure 3—figure supplement 1A, B*). A similar phenotype was

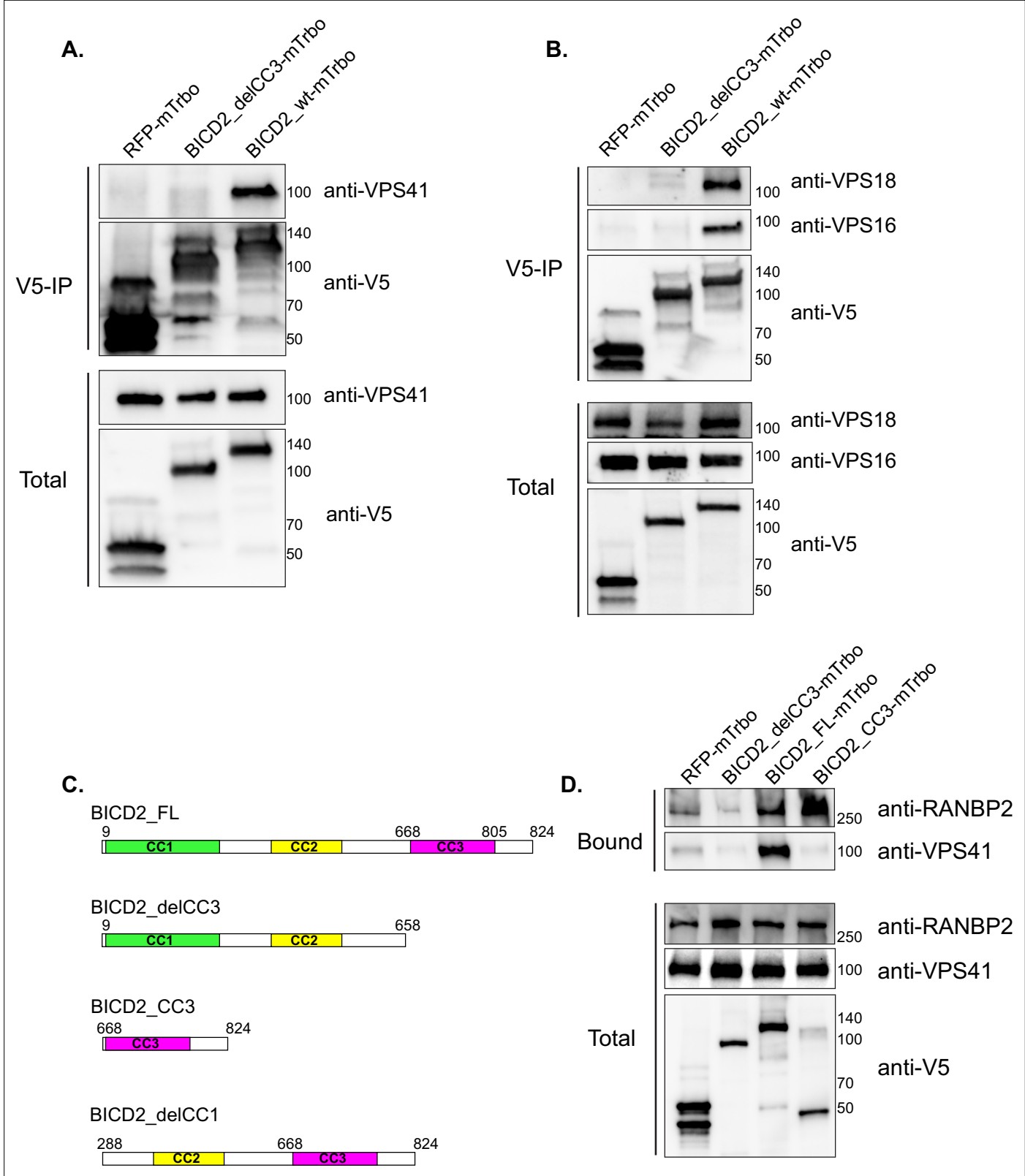

**Figure 2.** BICD2 interacts with components of the HOPS complex in vivo. (**A**) A co-immunoprecipitation experiment was performed with HEK cells expressing the indicated constructs. The lysates were incubated with V5 trap beads to precipitate the tagged proteins. The co-precipitating proteins were analyzed using western blotting with the indicated antibodies. VPS41 specifically co-precipitates with BICD2_wt. Minimal binding was observed with RFP-mTrbo or a BICD2 construct lacking the cargo binding domain. (**B**) A similar co-precipitation experiment was set up as in panel A. The co-

*Figure 2 continued*

precipitating proteins were analyzed by blotting using antibodies against VPS16, VPS18, and the V5 epitope. VPS16 and VPS18 co-precipitate specifically with BICD2_wt-mTrbo. (**C**) A schematic of the BICD2 constructs used in the binding experiment. (**D**) Cells expressing either RFP-mTrbo, BICD2_delCC3-mTrbo (lacking the cargo binding domain), BICD2_wt-mTrbo (full length BICD2) or BICD2_CC3-mTrbo (just the cargo binding domain) were used to examine the interaction with VPS41. Lysates were incubated with streptavidin beads to precipitate biotinylated proteins. The precipitated proteins were analyzed by blotting using the indicated antibodies. Although RANBP2 interacts with the isolated cargo binding domain of BICD2, VPS41 does not.

The online version of this article includes the following source data and figure supplement(s) for figure 2:

**Source data 1.** Original files for western blots displayed in *Figure 2A, B and D*.

**Source data 2.** PDF file for western blots displayed in *Figure 2A, B and D* with the bands marked.

**Figure supplement 1.** Characterization of the interaction between BICD2 and HOPS complex components.

**Figure supplement 1—source data 1.** Original files for western blots shown in *Figure 2—figure supplement 1A–D*.

**Figure supplement 1—source data 2.** PDF files for the western blots shown in *Figure 2—figure supplement 1A–D* with the bands marked.

also observed using a different siRNA targeting BICD2 (*Figure 3—figure supplement 1B and D*). Previous studies have shown that disruption of BICD2-mediated dynein cargo trafficking results in fragmentation of the Golgi and defective centrosome to nucleus tethering in cells in G2 phase of the cell cycle (*Hoogenraad et al., 2001*; *Splinter et al., 2010*). Consistent with these earlier findings, we also found that the depletion of BICD2 using the above siRNA resulted in Golgi fragmentation and centrosome localization defects in HeLa cells (*Figure 3—figure supplement 1E-H*).

In addition to the HOPS complex, late endosomes and lysosomes are also positive for LAMP1 and previous studies have shown partial co-localization between V5-tagged VPS41 and LAMP1 vesicles (*van der Welle et al., 2021*). We therefore used a validated antibody to detect LAMP1 vesicles under these same experimental conditions. Consistent with what was observed for GFP-VPS41, LAMP1 was present in more peripheral vesicles upon DHC depletion and was clustered closer to the nucleus in BICD2-depleted cells (*Figure 3F–H*, *Figure 3—figure supplement 1I–K*).

Although we were able to visualize GFP-VPS41 for immunofluorescence experiments using a GFP antibody to boost signal intensity, the native fluorescence of GFP-VPS41 was consistently very low, precluding our ability to examine the motility of these vesicles in living cells. We therefore examined lysosome motility using an SiR lysosome kit. This reagent detects cathepsin D present in mature lysosomes. As expected, although lysosomes were present throughout the cell, they were enriched in a perinuclear area, and a large fraction of these particles were motile (*Figure 3—video 1*). Depletion of BICD2 caused an even more pronounced perinuclear enrichment of lysosomes, consistent with the localization of GFP-VPS41 and LAMP1. However, in contrast to cells treated with the control siRNA, very few motile particles were detected in BICD2-depleted cells (*Figure 3—figure supplement 1L*, *Figure 3—video 2*). Thus, depletion of BICD2 results in perinuclear enriched lysosomes that are largely immotile.

The localization of GFP-VPS41 and LAMP1 vesicles in BICD2-depleted cells is similar to the phenotype obtained upon depletion of KIF5B, a plus-end directed Kinesin-1 motor (*Guardia et al., 2016*). Although BICD2 is primarily regarded as a cargo adaptor for dynein, it has also been shown to interact with KIF5B (*Grigoriev et al., 2007*). In addition, *Drosophila* BicD was recently shown to activate the motility of Kinesin-1, the fly ortholog of KIF5B (*Ali et al., 2025*). In order to more closely examine the role of BICD2 in LAMP1 vesicle localization, we overexpressed either GFP or BICD2 in cells. In contrast to cells overexpressing GFP, LAMP1 vesicles were peripherally scattered in BICD2 overexpressing cells. This phenotype was observed in HeLa and Cos7 cells (*Figure 3I, J, L and M*, *Figure 3—figure supplement 1M, M'*). Depletion of KIF5B in BICD2 overexpressing cells reverted this phenotype and restored the perinuclear localization of LAMP1 vesicles (*Figure 3K–M*, *Figure 3—figure supplement 1C*). These results suggest that BICD2 likely functions to link VPS41 and LAMP1 vesicles with KIF5B for plus-end directed transport. Our studies do not reveal the mechanism of this linkage, however, and further studies will be needed to fully understand the role of BICD2 in this process.

## SMALED2 mutations in the BICD2 cargo binding domain result in dynein hyperactivation

Several SMALED2-associated mutations have been identified in the BICD2 cargo binding domain (*Koboldt et al., 2020*). For this initial analysis, we chose to examine BICD2_R694C and BICD2_R747C.

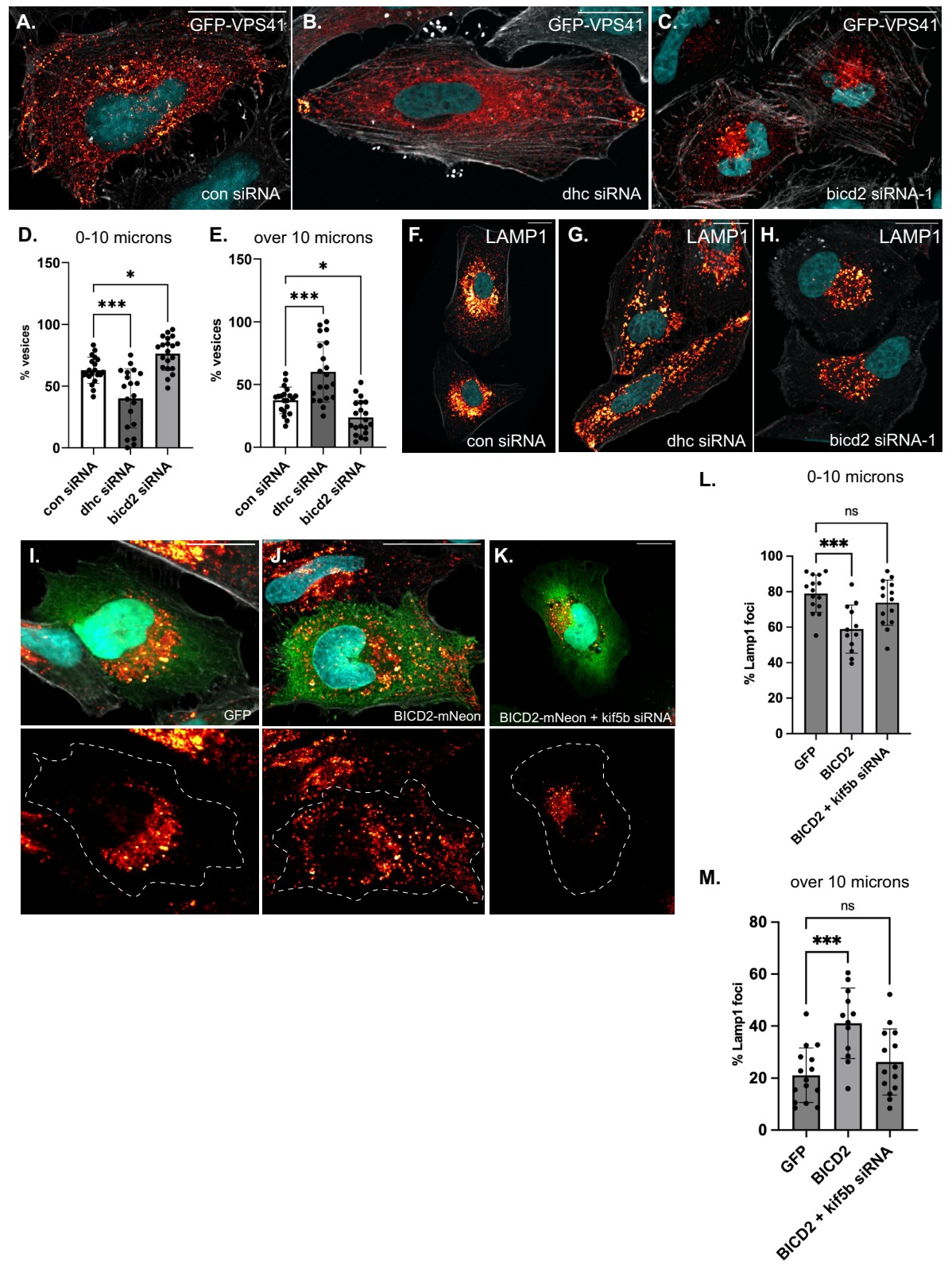

**Figure 3.** Role of BICD2 in localization of GFP-VPS41 and LAMP1 vesicles. (**A–C**) HeLa cells were transfected with either a control-siRNA (**A**), an siRNA targeting dynein heavy chain (**B**), or an siRNA targeting BICD2 (**C**). Two days after the siRNA transfection, the cells were transfected with a plasmid encoding GFP-VPS41. The next day, the cells were fixed and processed for immunofluorescence using an antibody against GFP. The cells were counterstained with DAPI (cyan) and Phalloidin (gray). Depletion of DHC results in an outward spreading of GFP-VPS41 vesicles, whereas depletion of

*Figure 3 continued on next page*

*Figure 3 continued*

BICD2 results in more perinuclear clustered vesicles. (**D, E**) The distance of GFP-VPS41 vesicles relative to the nucleus was determined and plotted. Vesicles present within 10 µm of the nucleus are shown in D, and those present at a distance greater than 10 µm are shown in panel E. (**F–H**) HeLa cells were transfected with the indicated siRNAs. Three days later, the cells were fixed and processed for immunofluorescence using an antibody against LAMP1. As with GFP-VPS41 vesicles, depletion of DHC resulted in peripheral vesicles, whereas depletion of BICD2 resulted in perinuclear clustering of LAMP1 vesicles. (**I–K**) HeLa cells were transfected with either a control siRNA (**I, J**) or an siRNA targeting KIF5B (**K**). Two days later, the cells were transfected with a plasmid encoding either GFP (**I**) or BICD2-mNeon (**J, K**). The cells were fixed on day 4 and processed for immunofluorescence using an antibody against LAMP1. The cells were counterstained with DAPI. Overexpression of BICD2 results in the peripheral spreading of LAMP1 vesicles. This phenotype was reversed upon knocking down KIF5B. (**L–M**) The distance of LAMP1 vesicles relative to the nucleus was determined and plotted. Vesicles present within 10 µm of the nucleus are shown in L, and those present at a distance greater than 10 µm are shown in panel M. The signal for GFP-VPS41 and LAMP1 is displayed using the 'red hot' LUT in FIJI. The scale bar is 20 µm. A one-way ANOVA was used for the quantifications shown in panels D, E, L, and M with the values compared to the mean of BICD2_wt. ns = not significant, *, $p \leq 0.05$, ***, $p < 0.001$.

The online version of this article includes the following video, source data, and figure supplement(s) for figure 3:

**Figure supplement 1.** Phenotypes associated with knock-down and over-expression of BICD2.

**Figure supplement 1—source data 1.** Original files for western blots displayed in *Figure 3—figure supplement 1A, B and C*.

**Figure supplement 1—source data 2.** PDF file for western blots displayed in *Figure 3—figure supplement 1A, B and C* with the bands marked.

**Figure 3—video 1.** Live imaging of SiR lysosome-labeled vesicles in HeLa cell transfected with a control siRNA.
https://elifesciences.org/articles/107503/figures#fig3video1

**Figure 3—video 2.** Live imaging of SiR lysosome-labeled vesicles in HeLa cell transfected with an siRNA targeting BICD2.
https://elifesciences.org/articles/107503/figures#fig3video2

These mutants, which result in the substitution of arginine for cystine at the indicated residues, were particularly intriguing because the R747C mutation was predicted to disrupt the interaction between BICD2 and RANBP2 (*Terawaki et al., 2015*). By contrast, previous studies suggest that the R694C mutation results in a higher level of interaction between BICD2 and RANBP2 (*Yi et al., 2023*). In addition to these mutations in the cargo binding domain, we also chose to analyze an additional mutant, BICD2_N188T (substitution of asparagine for threonine). This mutation is present within the first coiled coil domain and has been shown to result in dynein hyperactivation (*Huynh and Vale, 2017*). No studies have thus far addressed the global interactome of these mutants and how they might differ from the wild-type protein.

Stable cell lines capable of expressing BICD2_N188T, R694C, and R747C with a C-terminal mTrbo tag were generated in HEK293 FLP In T-Rex cells. We first determined whether the cargo binding domain mutants retained their ability to interact with dynein. Wild-type and mutant versions of BICD2-mTrbo, containing a V5 tag, were immunoprecipitated using V5 trap beads. The co-precipitated proteins were analyzed by western blotting using antibodies against Dynactin1/p150 Glued (DCTN1) and Dynein intermediate chain (DYNC1I1). Interestingly, not only did BICD2_R694C and BICD2_R747C retain their interaction with dynein, but both mutants also generally interacted with dynein and dynactin at a higher level than the wild-type protein (*Figure 4A*). Due to experimental variability in the co-immunoprecipitation experiment, the increased rate of binding was only statistically significant for the BICD2_R747C mutant (*Figure 4—figure supplement 1A and B*). In this regard, they were similar to the BICD2_N188T mutant which had previously been shown to interact with dynein at a higher level and to hyperactivate the motor (*Figure 4A*; *Huynh and Vale, 2017*).

We next examined the localization of wild-type and mutant BICD2 in HeLa cells. Wild-type BICD2 was localized throughout the cell with a fraction co-localizing with Golgi and centrosomal markers (*Figure 4B, F*, *Figure 4—figure supplement 1C, C' and C''*). By contrast, all three mutants were enriched at varying degrees within the centrosome (*Figure 4C–F*). Notably, the centrosomal enrichment was highly significant for BICD2_R694C and BICD2_R747C (*Figure 4F*). These results suggest that the cargo binding domain mutants might hyperactivate dynein. To test this more directly, we monitored the distribution of peroxisomes using a well-characterized tethering assay (*Kapitein et al., 2010*; *Passmore et al., 2021*). Peroxisomes are relatively immotile but can be transported towards microtubule minus ends in a dynein-dependent manner if they are tethered to an activating adaptor (*Figure 4G*). Thus, centrosomal clustering of peroxisomes is an indicator of dynein activity. Peroxisomes were tethered to wild-type or mutant alleles of BICD2. Consistent with the higher level of dynein interaction, peroxisomal clustering was increased in cells expressing all three SMALED2 mutants (*Figure 4B, F*, *Figure 4—figure supplement 1C, C' and C''*). Collectively, these results

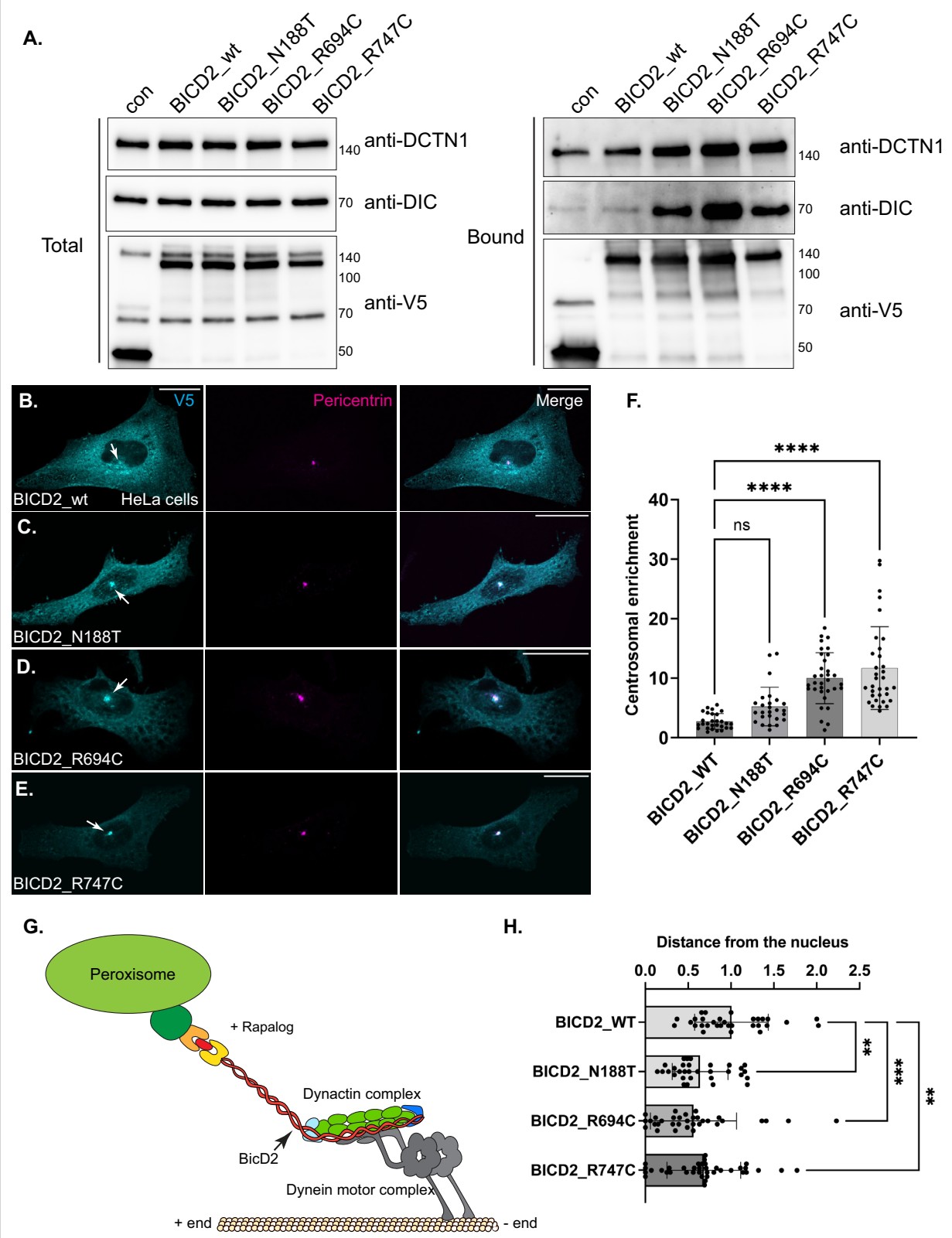

**Figure 4.** BICD2 cargo binding domain mutants hyperactivate dynein. (**A**) A co-immunoprecipitation experiment was performed using HEK cells expressing the indicated constructs. The tagged proteins were purified using V5 trap beads, and the co-precipitating proteins were analyzed by western blotting with the indicated antibodies. A greater amount of DIC and DCTN1 co-purified with mutant BICD2 compared to the wild-type protein. (**B–E**) HeLa cells expressing BICD2_wt (**B**), BICD2_N188T (**C**), BICD2_R694C (**D**), or BICD2_R747C (**E**) were fixed and processed for immunofluorescence

*Figure 4 continued on next page*

*Figure 4 continued*

using antibodies against V5 (cyan) and pericentrin (magenta). Merged images are also shown. All three mutants displayed a centrosomal localization pattern to varying degrees. The scale bar is 20 µm. (**F**) The centrosomal enrichment of BICD2_wt or mutant was quantified. (**G**) Schematic of the peroxisome tethering assay. (**H**) The average distance of peroxisomes to the nucleus was calculated on a cell-by-cell basis. In comparison to wild-type BICD2, all three mutants showed increased clustering of peroxisomes close to the nucleus. A one-way ANOVA was used for the quantifications shown in panels F and H with the values compared to the mean of BICD2_wt. n=not significant, **, p<0.01, ***, p<0.001, ****, p<0001.

The online version of this article includes the following source data and figure supplement(s) for figure 4:

**Source data 1.** Original files for western blots displayed in *Figure 4A*.

**Source data 2.** PDF file for western blots displayed in *Figure 4A* with the bands marked.

**Figure supplement 1.** BICD2 localization and interaction with dynein and KIF5B.

**Figure supplement 1—source data 1.** Original files for the western blot shown in *Figure 4—figure supplement 1H*.

**Figure supplement 1—source data 2.** Original files for the western blot shown in *Figure 4—figure supplement 1H* with bands marked.

suggest that SMALED2 mutations within the cargo binding domain of BICD2 are also capable of hyperactivating dynein.

As noted previously, BICD2 has also been shown to interact with the microtubule plus-end directed motor, KIF5B (*Grigoriev et al., 2007*). We therefore determined whether this interaction was maintained with the mutant alleles of BICD2. Wild-type full-length BICD2 associated with KIF5B at slightly higher levels than the binding control, consistent with published results (*Grigoriev et al., 2007*). Interestingly, all three mutants displayed a reduced interaction with KIF5B, with the BICD2_R747C mutant being the most severe (*Figure 4—figure supplement 1H and I*). Thus, the phenotypic outcome of dynein hyperactivity likely results from the combined effect of increased binding between BICD2 and dynein, and reduced binding between BICD2 and KIF5B.

In contrast to HeLa cells, microtubules are organized differently in neurons. In these cells, microtubule minus ends are localized within the cell body and plus ends extend towards the axon (*van Beuningen and Hoogenraad, 2016*). Thus, anterograde movement towards the axon tip is driven by a plus end motor, whereas retrograde transport towards the cell body involves dynein. Primary hippocampal neurons were established from E18 rat brains and after 3 days in vitro, plasmids containing either wild-type or mutant alleles of BICD2 were transfected into these cells. Consistent with BICD2 being a cargo adaptor for dynein, most of the signal for the wild-type protein was detected within the cell body (*Figure 5A*). However, wild-type BICD2 could also be detected within the axon. A similar pattern was noted for BICD2_N188T (*Figure 5B*). By contrast, axonal signal for BICD2_R694C and BICD2_R747C was reduced, reflected as a relative increase in cell body enrichment (*Figure 5C and D*). The difference in localization was statistically significant for the R747C mutant (*Figure 5E*). These results are consistent with the cargo binding domain mutants resulting in dynein hyperactivation. Huynh and Vale found that expression of certain SMALED2-associated BICD2 mutants, including BICD2_N188T, in hippocampal neurons results in reduced neurite growth (*Huynh and Vale, 2017*). We obtained similar results and found that this phenotype was also shared by the cargo binding domain mutations, BICD2_R694C and BICD2_R747C (*Figure 5F*).

## SMALED2 mutations result in altered BICD2 interactomes

We next determined the interactomes of the three SMALED2 mutations and compared these interactomes to that of wild-type BICD2. As before, the entire experiment was done in triplicate. Proteins that displayed a twofold increased or decreased interaction with the mutant allele in comparison to the wild-type and had a p value of at least 0.05 were considered significant (*Figure 6A–C*; *Supplementary files 2–4*). All three mutations, including N188T, which is present in the CC1 domain of BICD2, displayed an altered interactome. The most dramatic difference, however, was observed for BICD2_R747C (*Figure 6C*).

In order to validate the proteomics results, we repeated the experiment in triplicate and analyzed the pellets by western blotting. Consistent with the proteomics data, and with published results using a mutation in mouse BicD1 (*Terawaki et al., 2015*), BICD2_R747C displayed a reduced interaction with RANBP2 (*Figure 6D and E*; *Supplementary file 4*). BICD2 has been shown to associate indirectly with RANGAP via its direct interaction with RANBP2 (*Splinter et al., 2010*). Thus, as expected, a reduced level of RANGAP was detected in pellets from BICD2_R747C in comparison to the wild-type

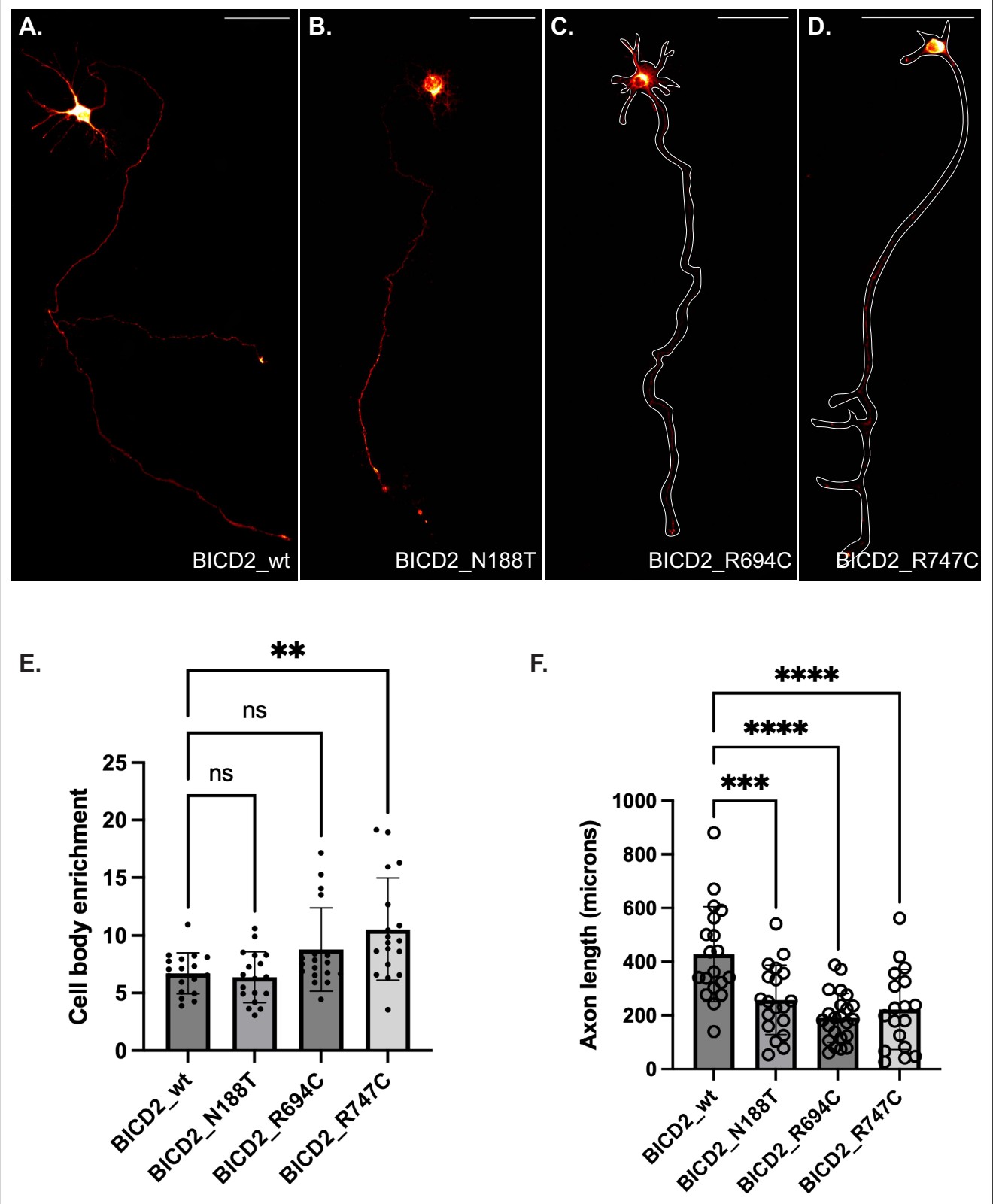

**Figure 5.** Localization of BICD2 wild-type and mutants in neurons. (**A–D**) E18 rat hippocampal neurons were transfected with the indicated constructs. Two days after transfection, the cells were fixed and processed for immunofluorescence using a V5 antibody. The axon outline for cells expressing BICD2_R694C and BICD2_R747C is indicated. Signal for wild-type BICD2 could be detected in the cell body and axon. A similar phenotype was noted for BICD2_N188T. By contrast, BICD2_R694C and BICD2_R747C displayed reduced axonal signal. The scale bar is 100 µm. The signal for BICD2 is

*Figure 5 continued on next page*

Figure 5 continued
displayed using the 'red hot' LUT in FIJI. (**E**) Quantification of the cell body enrichment of BICD2 wild-type and mutant. (**F**) The axon length of neurons expressing either wild-type or mutant alleles of BICD2 was quantified. Expression of BICD2 mutants correlated with shorter axonal lengths. A one-way ANOVA was used for the quantifications shown in panels E and F, and the values were compared to the mean of BICD2_wt. n=not significant, **, p<0.01, ***, p<0.001, ****, p<0.0001.

(*Supplementary file 4*). A further analysis of the proteomics results indicated a reduced association of BICD2_R747C with nuclear import receptors such as importin beta (*Supplementary file 4*). This result was confirmed by repeating the binding experiment and analyzing the eluate by western blotting (*Figure 6—figure supplement 1A and B*). We hypothesized that the association between BICD2 and import receptors is likely indirect and mediated via the direct interaction between BICD2 and RANBP2. Consistent with this notion, the interaction of wild-type BICD2 with importin beta was greatly reduced in cells depleted of RANBP2 (*Figure 6—figure supplement 1C*). By contrast, knock-down of RANBP2 had no effect on the BICD2-VPS41 interaction (*Figure 6—figure supplement 1C*). Based on these results, we conclude that BICD2_R747C is disrupted for interaction with RANBP2, and consequently also displays a reduced association with additional nuclear import factors.

An unexpected finding was that the BICD2_R747C mutant also displayed a reduced interaction with HOPS complex components. The proteomics result indicated a reduced interaction with VPS41, VPS18, and VPS33 (*Supplementary file 4*). However, the mass spectrometry result indicated that the association of BICD2_R747C with VPS16 was unaffected. In order to validate these results, the experiment was repeated and analyzed by western blotting. Indeed, the association of BICD2_R747C with VPS41 and VPS18 was greatly reduced in comparison to the wild-type protein (*Figure 6—figure supplement 1D*). However, using this approach, we consistently observed a significant reduction in the association between BICD2_R747C and VPS16 (*Figure 6—figure supplement 1D*). Based on these results, we conclude that this SMALED2 mutant is most likely compromised for interacting with the entire HOPS complex.

In addition to interactions that were lost or reduced, all three mutants were also associated with changes in the positive direction. For instance, the mass spectrometry results revealed that the mutants showed a stronger interaction with the centrosomal protein CSPP1 compared to wild-type BICD2 (*Supplementary files 2–4*). Validation experiments confirmed this finding, demonstrating that the cargo-binding domain mutants had significantly elevated interactions with CSPP1, exceeding that observed with either wild-type BICD2 or BICD2_N188T (*Figure 6D and G*). Since BICD2_R694C and BICD2_R747C are enriched at the centrosome (*Figure 4D–F*), their altered localization most likely accounts for their increased association with centrosomal proteins.

## BICD2_R747C exhibits a gain-of-function interaction with GRAMD1A

Among the three mutants analyzed, BICD2_R747C exhibited the most distinct interactome. One particularly notable interaction was with GRAMD1A, a protein absent from the wild-type BICD2 interactome but highly enriched in the BICD2_R747C pull-down, indicating a gain-of-function interaction. GRAMD1 encodes three isoforms: A, B, and C, with GRAMD1A being highly expressed in the CNS. To validate this observation, the experiment was repeated in triplicate and analyzed by western blotting. Consistent with the mass spectrometry result, GRAMD1A showed only background binding to the control, wild-type BICD2, BICD2_N188T, or BICD2_R694C. In contrast, GRAMD1A strongly associates with BICD2_R747C (*Figure 7A and B*). The same binding profile was observed in cells depleted of endogenous BICD2 (*Figure 7—figure supplement 1A*).

We next examined the localization of GRAMD1A in Cos7 cells co-expressing GRAMD1A-mScarlet3 with wild-type or mutant alleles of BICD2 tagged with mNeon. GRAMD1A localizes to the endoplasmic reticulum (ER) and concentrates at sites of ER-plasma membrane contact (*Besprozvannaya et al., 2018*). Cos7 cells are typically used for examining the localization of GRAMD1A due to their large and flattened appearance (*Besprozvannaya et al., 2018*; *Naito et al., 2019*). As expected, GRAMD1A displayed a typical ER localization pattern in cells expressing wild-type BICD2 or BicD2_N188T (*Figure 7C and D*). Much like what was observed in HeLa cells, BICD2_R694C was often enriched in the centrosomal region in Cos7 cells (*Figures 7E and 4D*). The localization of GRAMD1A was essentially unchanged in these cells, although a small amount of the protein co-localized with BICD2_R694C in the centrosomal region (*Figure 7E and G*). By contrast, consistent with a gain-of-function interaction,

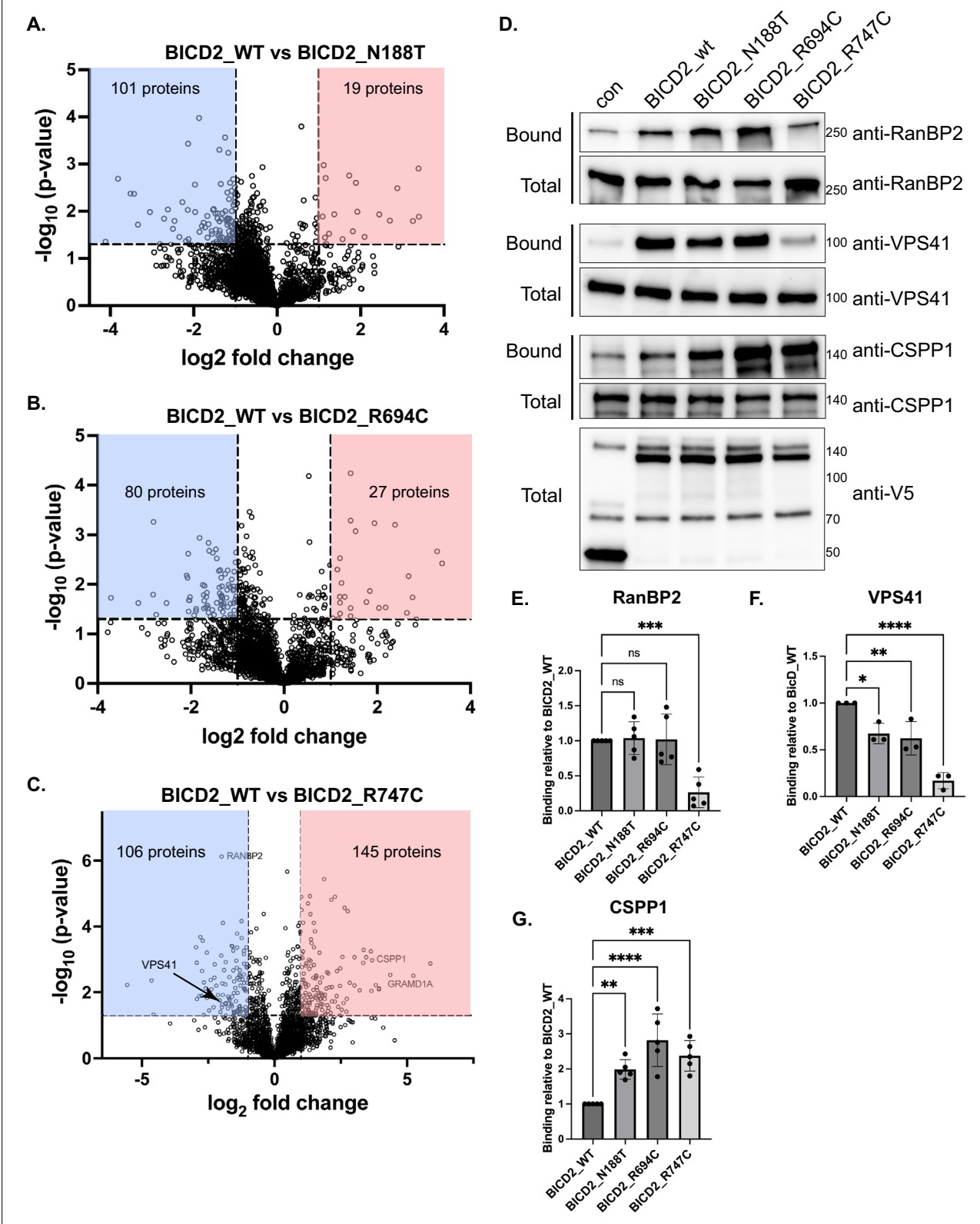

**Figure 6.** BICD2 mutations are associated with altered interactomes. (**A–C**). Volcano plots comparing the interactome of BICD2_wt vs BICD2_N188T (**A**), vs BICD2_R694C (**B**) and vs BICD2_R747C (**C**). Interacting proteins that show at least a twofold change in comparison to BICD2_wt and have a p value of at least 0.05 are indicated in the shaded boxes. Red boxes indicate proteins that display a greater interaction with BICD2 mutants vs the wild-type, whereas blue boxes indicate proteins that display a lower interaction vs the wild-type protein. (**D**) The proteomics results were validated by

*Figure 6 continued on next page*

*Figure 6 continued*

repeating the experiment and analyzing the bound fractions using the indicated antibodies. Streptavidin beads were used to purify the biotinylated proteins. (**E–G**) Quantification of binding of BICD2_wt and mutants with RANBP2 (**E**), VPS41 (**F**), and CSPP1 (**G**). The level of binding for the mutants was compared to BICD2_wt. Consistent with the proteomics results, BICD2_R747C displayed reduced binding to RANBP2 and VPS41. All three mutants bound CSPP1 at a greater level than the wild-type. A one-way ANOVA was used for this analysis. ns = not significant, *, $p \leq 0.05$, **, $p < 0.01$, ***, $p < 0.001$, ****, $p < 0.0001$.

The online version of this article includes the following source data and figure supplement(s) for figure 6:

**Source data 1.** Original files for western blots displayed in *Figure 6D*.

**Source data 2.** PDF file for western blots displayed in *Figure 6D* with the bands marked.

**Figure supplement 1.** BICD2 interaction with nuclear import receptors and the HOPS complex.

**Figure supplement 1—source data 1.** Original files for western blots shown in *Figure 6—figure supplement 1A-D*.

**Figure supplement 1—source data 2.** Original files for western blots shown in *Figure 6—figure supplement 1A-D* with bands indicated.

a significant fraction of GRAMD1A localized adjacent to Pericentrin, a centrosomal marker, in cells expressing BICD2_R747C (*Figure 7F*; *Figure 7—figure supplement 1B–F*). This effect was specific for GRAMD1A because a general ER marker remained correctly localized in cells expressing BICD2_R747C (*Figure 7—figure supplement 1G, G', H and H'*).

Collectively, our results suggest that SMALED2-associated mutations in BICD2 hyperactive dynein and result in gain-of-function and loss-of-function interactions with cargo. It is therefore possible that cargo trafficking defects that arise because of these phenotypes contribute to the symptoms associated with SMALED2.

## Discussion

Mutations in the dynein cargo adaptor BICD2 have been linked to SMALED2 (*Koboldt et al., 2020*). Mutations in the heavy chain of the dynein motor have also been implicated in a version of this disorder (*Chan et al., 2018*; *Das et al., 2018*), suggesting that defects in dynein-mediated transport contribute to its etiology. However, the molecular and cellular mechanisms underlying SMALED2 pathogenesis remain poorly understood. Previous studies have characterized mutations within the first coiled-coil domain of BICD2, a region responsible for interactions with dynein and dynactin. These analyses elegantly demonstrated that mutants such as BICD2_N188T result in dynein hyperactivity (*Huynh and Vale, 2017*). In addition to these mutants, however, recent studies have identified several SMALED2-associated alleles within the C-terminal cargo-binding domain of BICD2 (*Ravenscroft et al., 2016*; *Synofzik et al., 2014*). Given BICD2's role as a dynein cargo adaptor, these findings raise two important questions: (1) Is dynein hyperactivity a common feature of SMALED2-associated BICD2 mutations? and (2) Do these mutations alter the interactome of BICD2 relative to the wild-type protein? The goal of this study was to address these questions and elucidate potential molecular consequences of SMALED2-associated BICD2 mutations.

BICD2 is one of the best characterized dynein cargo adaptors. However, most studies involving BICD2 have focused on the mechanism by which this adaptor activates dynein for processive motility. Relatively little is known regarding the cargo that is linked to dynein by BICD2. In *Drosophila*, BicD links the RNA-binding protein Egalitarian (Egl) with dynein for transport of specific mRNAs in the oocyte and embryo (*Dienstbier et al., 2009*; *Goldman et al., 2021*; *Goldman et al., 2019*; *Mach and Lehmann, 1997*; *McClintock et al., 2018*). Loss of either BicD or Egl compromises transport of these mRNAs and consequently results in defective oogenesis or embryogenesis. The first definitive cargo identified for mammalian BICD2 was the small GTP-binding protein, RAB6A (*Matanis et al., 2002*). Despite the ability of BICD2 to directly bind RAB6A, most vesicles containing RAB6A move towards the plus end of microtubules, suggesting that their transport is primarily driven by the Kinesin-1 motor, KIF5B (*Grigoriev et al., 2007*). Other cargos that have been shown to directly bind BICD2 are RANBP2, a nucleoporin, and Nesprin-2 (SYNE2), a LINC complex component involved in linking dynein and kinesin to the nuclear envelope (*Gonçalves et al., 2020*; *Splinter et al., 2010*).

In order to determine whether SMALED2 alleles of BICD2 are associated with interactome changes, it was therefore critical for us to determine the interactome of wild-type BICD2. This was done using the promiscuous biotin ligase miniTurboID (mTrbo). In comparison to an RFP-mTrbo

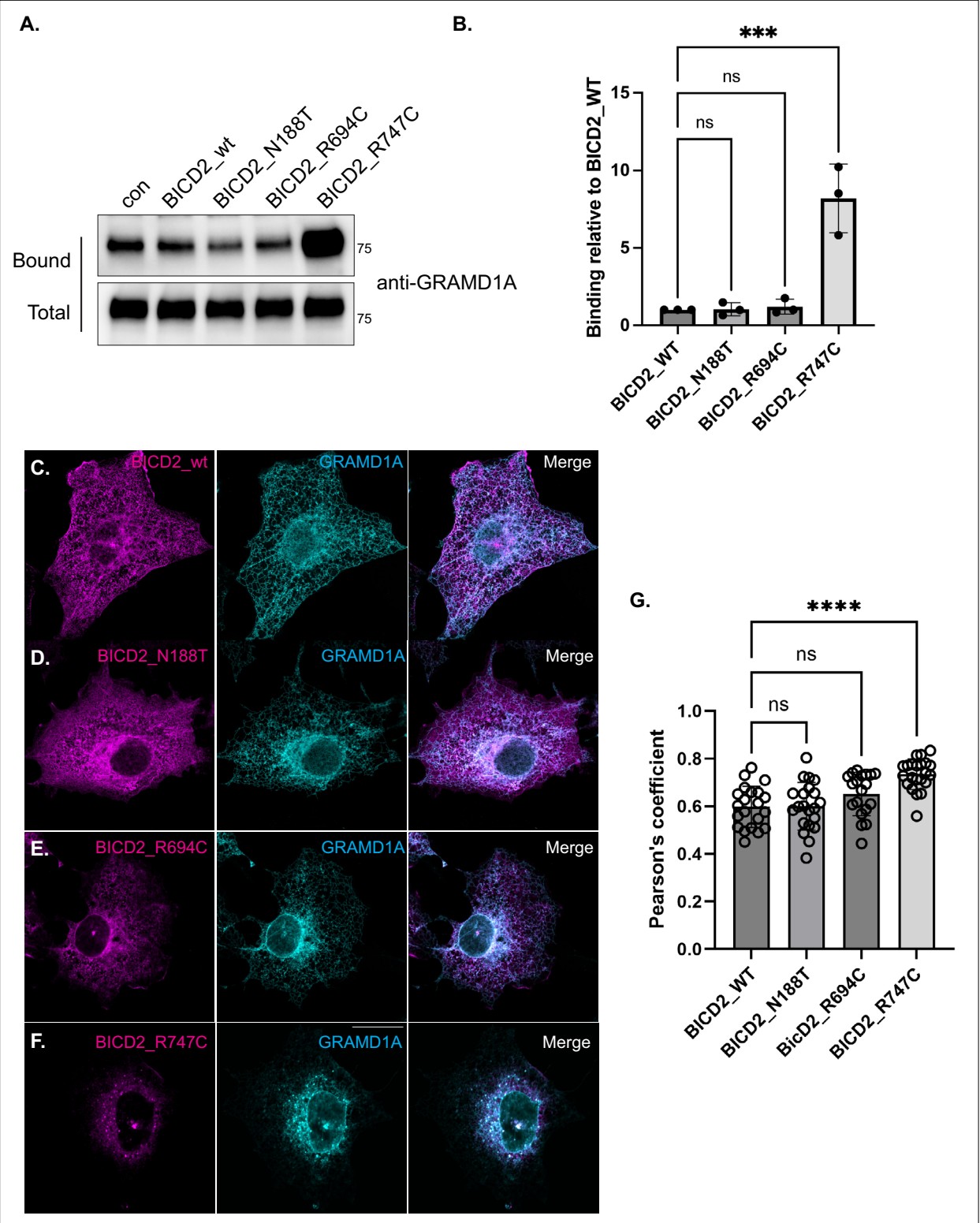

**Figure 7.** BICD2_R747C is associated with a gain-of-function interaction with GRAMD1A. (**A**) Lysates from cells expressing BICD2_wt and mutants were incubated with streptavidin beads to purify biotinylated proteins. Bound proteins were eluted and analyzed by blotting using the indicated antibodies. BICD2_R747C interacted with substantially more GRAMD1A than either the control, BICD2_wt, or the other mutants. (**B**) Quantification of binding of BICD2_wt and mutants with GRAMD1A. The level of binding for the mutants was normalized to BICD2_wt. A one-way ANOVA was used for this analysis. ns = not significant, \*\*\*, p<0.001. (**C–F**) Cos7 cells were co-transfected with constructs expressing either BICD2_wt or mutant (magenta) along with a plasmid expressing GRAMD1A-mScarlet3 (cyan). Except for cells expressing BICD2_R747C, GRAMD1A was localized to the ER. By contrast, GRAMD1A

*Figure 7 continued on next page*

*Figure 7 continued*

was highly enriched at the centrosome in cells expressing BICD2_R747C. The scale bar is 20 µm. (G) Quantification of the co-localization between BICD2_wt and mutants with GRAMD1A. A one-way ANOVA was used for this analysis and the values were compared to the mean of BICD2_wt. ns = not significant, ****, p<0.0001.

The online version of this article includes the following source data and figure supplement(s) for figure 7:

**Source data 1.** Original files for western blots displayed in *Figure 7A* .

**Source data 2.** PDF file for western blots displayed in *Figure 7A* with the bands marked.

**Figure supplement 1.** BICD2 interaction and co-localization with GRAMD1A.

**Figure supplement 1—source data 1.** Original data for western blots shown in *Figure 7—figure supplement 1A*.

**Figure supplement 1—source data 2.** Original data for western blots shown in *Figure 7—figure supplement 1A* with bands indicated.

control, BICD2-mTrbo resulted in the biotinylation and purification of numerous known interacting partners including RANBP2, as well as several components of the dynein motor. One interesting group of potentially novel interacting proteins was components of the HOPS complex, a six-subunit complex of proteins involved in endocytic trafficking (*Spang, 2016*). Four of the six HOPS components were identified in the wild-type BICD2 interactome, with VPS41 being the fifth most enriched protein. However, unlike RANBP2, RAB6A, and NESPRIN-2, all of which are able to bind the isolated BICD2 cargo binding domain (*Gonçalves et al., 2020*; *Matanis et al., 2002*; *Splinter et al., 2010*), the HOPS complex components were only able to bind full-length BICD2. The BICD2 cargo binding domain was therefore necessary but not sufficient for interaction with HOPS components. In addition, contrary to our initial hypothesis that VPS41 was the direct binding partner between BICD2 and the HOPS complex, BICD2 retained its interaction with VPS16 and VPS18 in cells depleted of VPS41. This suggests that BICD2 likely recognizes a domain or motif present in several HOPS proteins. We attempted to use Alphafold2 multimer to predict the relevant domain within HOPS proteins that interact with BICD2. Although Alphafold2 was able to generate a high confidence prediction of the interaction site between BICD2 and RAB6A, consistent with published results (*Zhao et al., 2024*), it failed to produce a high confidence prediction for the BICD2-HOPS complex interaction (data not shown). Thus, although we were able to validate the in vivo association between BICD2 and VPS41, VPS16, and VPS18, we are not able to conclude whether BICD2 is capable of directly interacting with these proteins. To the best of our knowledge, this is the first example of BICD2 interacting proteins that display this binding characteristic. The ScaC protein from the intracellular pathogen *Orientia tsutsugamushi* was recently also shown to interact with BICD2, and although the binding site of ScaC was different from that used by RANBP2 or RAB6A, it was still able to interact with the isolated cargo binding domain of BICD2 (*Manigrasso et al., 2025*).

Another unusual aspect of the BICD2-HOPS complex interaction is that it does not appear to be linked to dynein-mediated trafficking. Depletion of dynein heavy chain resulted in the peripheral distribution of GFP-VPS41 and LAMP1 vesicles, indicative of a reduction in minus end transport, and a net gain in plus end directed transport. By contrast, depletion of BICD2 resulted in the perinuclear accumulation of lysosomal vesicles that were mostly immotile. Interestingly, however, overexpression of BICD2 caused the outward spreading of LAMP1 vesicles, a process that depends on KIF5B (*Guardia et al., 2016*). Previous studies have shown that BICD2 is also able to interact with KIF5B via a central coiled coil domain (*Grigoriev et al., 2007*; *Hoogenraad and Akhmanova, 2016*). A recent report suggests that *Drosophila* BicD is capable of interacting with and activating the motility of Kinesin-1, the fly homolog of KIF5B (*Ali et al., 2025*). Consistent with the notion that BICD2 might link late endosomal vesicles with KIF5B, depletion of KIF5B in BICD2 overexpressing cells restored the normal localization of LAMP1 vesicles. Additional studies will be required to determine whether BICD2 is capable of directly interacting with these vesicles and whether these vesicles are directly linked to KIF5B by BICD2.

The motility of LAMP1 vesicles has some similarity to the transport of RAB6A exocytic vesicles. RAB6A vesicles are transported from the area of the Golgi towards the cell periphery in a KIF5B-dependent manner, and loss of either kinesin-1 or dynein results in a sharp reduction in the number of motile particles (*Grigoriev et al., 2007*). In addition, mutations in BICD2 that compromise binding to RAB6A also result in vesicles that are largely immotile (*Zhao et al., 2024*). Thus, in the case of LAMP1 and RAB6A vesicles, instead of resulting in an increased rate of minus end transport, loss of BICD2

results in compromised vesicle motility, indicating that coordination between opposite polarity motors is critical for their motility.

As noted earlier, mutations in the CC1 region of BICD2 hyperactivate dynein (*Huynh and Vale, 2017*). Our findings indicate that this property is also shared by BICD2_R694C and BICD2_R747C, mutations present within the C-terminal cargo binding domain. In the absence of cargo, BICD2 is thought to exist in an inhibited conformation due to intramolecular interactions between the N and C termini of the protein (*Figure 1B*; *Liu et al., 2013*; *Terawaki et al., 2015*; *Wharton and Struhl, 1989*). Cargo binding to the C-terminus of BICD2 counteracts the intramolecular interaction, enabling N-terminal residues within BICD2 to bind the dynein/dynactin complex (*Goldman et al., 2019*; *Huynh and Vale, 2017*; *Liu et al., 2013*; *McClintock et al., 2018*; *Sladewski et al., 2018*). How might mutations in BICD2 result in dynein hyperactivation? One possibility is that these mutations disrupt the autoinhibited state of BICD2, effectively causing BICD2 to be present in a more open and uninhibited conformation that promotes dynein/dynactin binding. Molecular dynamics simulations suggest that the R747C substitution causes a registry shift in the coiled coil, likely destabilizing this domain and thus disrupting the intramolecular interaction between the N and C termini of BICD2 (*Cui et al., 2020*). Another possibility is that the hyperactivation of dynein results in reduced binding between BICD2 and KIF5B. Our results are consistent with this scenario and suggest that the net effect of dynein hyperactivity results in three molecular changes; reduced intramolecular BICD2 interaction, increased interaction between BICD2 and dynein, and reduced interaction between BICD2 and KIF5B.

In addition to hyperactivating dynein, all three mutations, including BICD2_N188T, alter the BICD2 interactome. This finding was unexpected for BICD2_N188T because this mutation is not within the cargo binding domain. One possible explanation for this phenotype is that BICD2_N188T is present in a more open conformation, and this change affects its binding properties. Another possibility that is not mutually exclusive is that the different binding profile results from the altered localization of BICD2_N188T within the cell. In comparison to wild-type BICD2, we generally observed greater centrosomal enrichment of BICD2_N188T. In comparing the three mutants, the general trend was that more proteins displayed a reduced interaction with the SMALED2 mutants in comparison to wild-type BICD2. Among the three mutants analyzed, BICD2_R747C displayed the most drastically altered interactome. This mutant displayed reduced association with RANBP2, importin beta, and HOPS complex components. Interestingly, this mutant also displayed numerous gain-of-function interactions. For instance, although minimal binding was observed between wild-type BICD2 and GRAMD1A, this protein abundantly interacted with BICD2_R747C. GRAMD1A is involved in nonvesicular transport of accessible cholesterol from the plasma membrane to the ER and is often concentrated at sites of plasma membrane-ER contact (*Besprozvannaya et al., 2018*; *Sandhu et al., 2018*). However, in cells expressing BICD2_R747C, this localization pattern was disrupted and GRAMD1A co-localized with BICD2_R747C adjacent to the centrosome.

The GRAMD1 family consists of three isoforms: GRAMD1A, GRAMD1B, and GRAMD1C. Interestingly, our interactome analysis only identified GRAMD1A as a gain-of-function interaction partner with BICD2_R747C. It is unclear whether GRAMD1B and GRAMD1C also interact with BICD2_R747C. However, given that GRAMD1 proteins can form hetero oligomers (*Naito et al., 2019*), the BICD2_R747C-induced mislocalization of GRAMD1A could potentially affect the distribution of other GRAMD1 isoforms as well. The GRAMD1 proteins function to sense excess accessible cholesterol in the plasma membrane and to mediate the transport of this cholesterol to the ER. This reduces the rate of new cholesterol synthesis by the ER, enabling the cell to maintain cholesterol homeostasis (*Sandhu et al., 2018*). It will be interesting to determine whether endogenous GRAMD1A is mislocalized in motor neurons of SMALED2 patients with the BICD2_R747C mutation, and if this results in an expanded accessible pool of cholesterol at the plasma membrane.

A recent study by Yi and colleagues examined the effect of the BICD2_R694C mutation on cargo binding (*Yi et al., 2023*). Using in vitro experiments, they found that this mutation enhanced RANBP2 binding while having no effect on NESPRIN-2 binding (*Yi et al., 2023*). Our results using full-length BICD2 are consistent with this finding. We also observed slightly higher binding of BICD2_R694C to RANBP2. However, due to experimental variability, the increase was not statistically significant. The authors also examined cargo binding using a BICD2 double mutant (F743I/R747C). Consistent with our results, this mutant displayed greatly reduced binding to RANBP2, but bound NESPRIN-2 at a much higher level than the wild-type protein (*Yi et al., 2023*). NESPRIN-2 was not identified as an

interacting partner in our study for the wild-type protein or the BICD2_R747C mutant, possibly due to its low expression level in HEK293 cells. Nevertheless, these findings, along with our interactome analysis, indicate that mutations in the cargo binding domain of BICD2 can result in loss- and gain-of-function interactions.

In conclusion, our study is the first to comprehensively examine the interactome of wild-type BICD2 and to identify changes that occur in SMALED2 linked mutant alleles of BICD2. We find that not only are mutations within the cargo binding domain associated with interactome changes, but these mutations are also capable of hyperactivating dynein. Some limitations of this study are worth noting. In the current study, we chose to determine the BICD2 interactome in HEK FLP-In cells (embryonic kidney cells). These cells were chosen because they enabled us to precisely integrate wild-type and mutant alleles of BICD2 at a specific locus. It also enabled us to expand cultures of these cells to levels that were sufficient for proteomic analysis. However, the main cell type affected in patients with SMALED2 is motor neurons. Primary motor neurons are harder to culture to scale and to genetically manipulate to express the desired wild-type or mutant BICD2 transgenes. Thus, although motor neurons were not used in our study, the next significant challenge will be to perform these types of experiments using motor neurons. In addition, although our study identified interactome changes between wild-type and mutant alleles of BICD2, we cannot conclude whether these changes are causative for the symptoms associated with SMALED2. Patients diagnosed with this disorder display a range of symptoms, from relatively mild to more severe (*Frasquet et al., 2020*; *Koboldt et al., 2020*). Even patients with the same genetic mutation can display a range of phenotypes (*Storbeck et al., 2017*). Furthermore, disease symptoms can result from one or two interactome changes that are critical for the health of motor neurons. Alternatively, symptoms might also be caused by many small changes in the interactome that cumulatively affect the health of motor neurons. Lastly, because SMALED2 is an autosomal dominant disorder, patients express wild-type and mutant versions of BICD2 in the same cell. Thus, to accurately model this disorder, studies will need to be conducted in motor neurons that are genetically edited to express disease-associated mutations in a heterozygous state.

## Materials and methods
### DNA constructs

Synthesized DNA fragments were generated by either Twist biosciences or Genewiz/Azenta. All final plasmids used in this work were sequenced by Plasmidsaurus. Plasmid sequences as well as detailed cloning strategies will be provided upon request.

DNA encoding wild-type BICD2, generated by gene synthesis, was cloned into the Kpn1 and Xho1 sites of the pCDNA-FRT-TO vector (Thermo Fisher Scientific). Next, a fragment encoding mTrbo was cloned downstream of BICD2_wild-type using high-fidelity assembly (NEB). A similar strategy using a synthesized fragment was used to clone mRFP-mTrbo into the pCDNA-FRT-TO vector. BICD2 mutants fused to mTrbo were generated by swapping in a synthesized fragment containing the desired mutation into the appropriate site of the BICD2_wild-type-mTrbo construct. The cDNA for wild-type BICD2 as well as the SMALED2 mutants contained silent mutations that made them resistant to bicd2 siRNA-1. The BICD2 truncations were produced by cloning either PCR-generated or gene-synthesized fragments into the appropriate vector. Stable cell lines were created using these FRT constructs and the pOG44 vector (Thermo Fisher Scientific). For localization studies, PCR sub-cloning was used to move the appropriate wild-type or BICD2 mutants into the pLVX-Neo vector (Takara Bio). A fragment encoding mNeon-green along with the V5 epitope tag was cloned downstream of wild-type or mutant BICD2. The GFP-VPS41 construct was generated by cloning a synthesized fragment encoding human VPS41 into the Kpn1 and Not1 sites of the GFP-BICD2 plasmid (Addgene plasmid #161626; *Bonet-Ponce et al., 2020*). This removed the BICD2 cDNA from this vector and inserted the cDNA for VPS41 in its place. For the dynein activity experiment described in *Figure 4*, a construct containing the PEX3 peroxisome targeting sequence fused to mGreenLantern and the FRB dimerization sequence was cloned into the pLVX neo vector (Takara). The BICD2_wild type and mutant construct described above were modified by replacing the mTrbo sequence with the FKBP dimerization domain. The GRAMD1A-mScarlet3 construct was generated by cloning fragments corresponding to human GRAMD1A and mScarlet3 into the pLVX neo vector.

## Cell culture

HeLa and Cos7 cells were authenticated and obtained from ATCC. These cells were cultured in Dulbecco's Modified Eagle Medium (DMEM) supplemented with 10% fetal bovine serum and 1% penicillin/streptomycin (Thermo Fisher Scientific). The HEK Flp-In T-REx 293 Cell Line was obtained from Thermo Fisher Scientific and was cultured according to the instructions provided by the manufacturer. HEK Flp-In T-REx 293 cells were authenticated by their resistance to Blasticidin and Zeocin and for the presence of the FRT recombination site by genomic DNA PCR. Stable cells were generated by culturing the transfected HEK Flp-In T-REx 293 in 100 µg/ml Hygromycin B (Gibco) for 12–14 days, with fresh media changes every 3 days. The E18 rat hippocampus culturing kit was obtained from Transnetyx. Primary neurons were prepared according to the instructions provided by the manufacturer. 40,000 cells were seeded onto poly-Lysine-coated coverslips (Neuvitro Corporation) in each well of a 12-well plate. The cells were transfected after 3–4 days in culture using Lipofectamine 2000 (Thermo Fisher Scientific).

Stable HEK Flp-In T-REx 293 expressing mGreenLantern tagged peroxisomes were generated by infection with lentivirus. Lentiviral particles were produced in HEK293T cells following a previously published protocol (*Wei et al., 2024*). Culture medium containing the viral particles was passed through a 0.2 µm filter to remove cell debris. Next, the HEK Flp-In T-REx 293 cells were infected with freshly prepared viral particles for 48 hr. After infection, puromycin (1 µg/mL) was added to the culture medium to select for stable cell lines. Uniformly expressing mGreenLantern clones were selected and used to integrate BICD2_wt or mutant containing the FKBP dimerization motif at the FRT locus. Stable cells containing the integrated BICD2 constructs were selected using Hygromycin B as described above.

The Universal Mycoplasma Detection Kit from ATCC was used to verify that the cell lines used in these experiments were free from mycoplasma contamination.

## DNA and siRNA transfections

The Qiagen Effectene reagent was used for transfecting DNA into HeLa, HEK Flp-In T-REx 293, and Cos7 cells using the directions provided by the manufacturer. For transfecting cells in six-well dishes, 0.4 µg of DNA was used along with 10 µl of Effectene. Primary neurons grown on glass coverslips in a 12-well dish were transfected using Lipofectamine 2000 (Thermo Fisher Scientific). 0.5–0.8 µg of DNA and 2 µl of Lipofectamine 2000 was used in each transfection. Expression of the transgenes was induced using 0.3 µg/ml of Doxycycline 24 hr after transfection. The cells were fixed and processed for immunofluorescence the following day. Lipofectamine RNAimax (Thermo Fisher Scientific) was used for the transfection of siRNA according to the instructions provided by the manufacturer.

## Purification and analysis of biotinylated proteins

For small scale experiments, HEK Flp-In T-REx 293 cells expressing the desired constructs by Tet-based induction (1 µg/ml, 24 hr) were incubated with biotin (500 µM) for 40 min. The cells were then harvested, and lysates were prepared by resuspending the cells in RIPA buffer (50 mM Tris-Cl [pH 7.5], 150 mM NaCl, 1% NP-40, 1 mM EDTA) containing a Halt Protease inhibitor cocktail (Thermo Fisher Scientific). 1 mg of total protein was used in the binding experiment with 15 µl of High-Capacity Streptavidin Agarose beads (Thermo Fisher Scientific) in RIPA buffer. The binding was performed overnight at 4 °C. The samples were washed four times using RIPA buffer, bound proteins were eluted in Laemmli buffer and analyzed by western blotting. All western blot images were acquired on a Bio Rad ChemiDoc MP.

For proteomics experiments, the same cells were grown in 10 cm dishes and 5 mg of total protein was used in each purification. Biotinylated proteins were purified using 70 µl of Streptavidin magnetic beads (Thermo Fisher Scientific), incubated at 4 °C overnight. The following day, the samples were extensively washed using 1 ml of the following: three times with RIPA buffer, three times with high-salt RIPA buffer (50 mM Tris-Cl [pH 7.5], 1 M NaCl, 1% NP-40, 1 mM EDTA), three times with RIPA buffer, and four times with PBS. The entire experiment was done in triplicate. After the final wash, the beads were resuspended in 70 µl of PBS and shipped to the Emory Integrated Proteomics Core on dry ice.

## Mass spectrometry

The mass spectrometry was performed at the Emory Integrated Proteomics Core (RRID:SCR_023530).

## On-bead digestion

A published protocol was followed for on-bead digestion of proteins (*Soucek et al., 2016*). A digestion buffer containing 50 mM $NH_4HCO_3$ was added to the beads. The mixture was then incubated with 1 mM dithiothreitol (DTT) at room temperature for 30 min, followed by the addition of 5 mM iodoacetimide (IAA). This mixture was incubated at room temperature for an additional 30 min in the dark. Proteins were digested with 1 µg of lysyl endopeptidase (Wako) at room temperature overnight and were further digested overnight at room temperature with 1 µg trypsin (Promega). The resulting peptides were desalted using an HLB column (Waters) and were dried under vacuum.

## LC-MS/MS

The data acquisition by LC-MS/MS was adapted from a published procedure (*Seyfried et al., 2017*). Derived peptides were resuspended in the loading buffer (0.1% trifluoroacetic acid, TFA) and were separated on a Water's Charged Surface Hybrid (CSH) column (150 µm internal diameter (ID) x 15 cm; particle size: 1.7 µm). The samples were run on an EVOSEP liquid chromatography system using the 15 samples per day preset gradient and were monitored on a Q-Exactive Plus Hybrid Quadrupole-Orbitrap Mass Spectrometer (Thermo Fisher Scientific). The mass spectrometer cycle was programmed to collect one full MS scan followed by 20 data-dependent MS/MS scans. The MS scans (400–1600 m/z range, $3x10^6$ AGC target, 100ms maximum ion time) were collected at a resolution of 70,000 at m/z 200 in profile mode. The HCD MS/MS spectra (1.6 m/z isolation width, 28% collision energy, $1x10^5$ AGC target, 100 ms maximum ion time) were acquired at a resolution of 17,500 at m/z 200. Dynamic exclusion was set to exclude previously sequenced precursor ions for 30 seconds. Precursor ions with +1, and +7, +8, or higher charge states were excluded from sequencing.

## MaxQuant

Label-free quantification analysis was adapted from a published procedure (*Seyfried et al., 2017*). Spectra were searched using the search engine Andromeda, integrated into MaxQuant, against 2022 human UniProtKB/Swiss-Prot database (20,387 target sequences). Methionine oxidation (+15.9949 Da), asparagine and glutamine deamidation (+0.9840 Da), and protein N-terminal acetylation (+42.0106 Da) were variable modifications (up to 5 allowed per peptide); cysteine was assigned as a fixed carbamidomethyl modification (+57.0215 Da). Only fully tryptic peptides were considered with up to 2 missed cleavages in the database search. A precursor mass tolerance of ±20 ppm was applied prior to mass accuracy calibration and ±4.5 ppm after internal MaxQuant calibration. Other search settings included a maximum peptide mass of 6,000 Da, a minimum peptide length of 6 residues, 0.05 Da tolerance for orbitrap and 0.6 Da tolerance for ion trap MS/MS scans. The false discovery rate (FDR) for peptide spectral matches, proteins, and site decoy fraction was all set to 1%.

## Quantification settings were as follows

Re-quantify with a second peak finding attempt after protein identification has completed; match MS1 peaks between runs; a 0.7 min retention time match window was used after an alignment function was found with a 20-min RT search space. Quantitation of proteins was performed using summed peptide intensities given by MaxQuant. The quantitation method only considered razor plus unique peptides for protein level quantitation.

## Co-immunoprecipitation

HEK293 FLP-in T-Rex cells expressing the desired constructs by Tet-based induction (1 µg/ml, 24 hr) were harvested, and lysates were prepared using RIPA buffer. 1 mg of total protein was used in the binding experiment with 15 µl of V5-Trap agarose beads (ChromoTek, ProteinTech) in RIPA buffer. The binding was performed at 4 °C for 2 hr. The samples were then washed four times using RIPA buffer, bound proteins were eluted in Laemmli buffer and analyzed by western blotting using the indicated antibodies.

## Immunofluorescence

Cells (HeLa, HEK293 FLP In T-Rex, Cos7, and primary rat neurons) adhered to glass coverslips were fixed using either 4% formaldehyde for 5 mins at room temperature (*Figures 5 and 7*) or with methanol

at –20 °C for 10 min (*Figures 3 and 4*). After fixation, cells were permeabilized by incubating with PBST (PBS containing 0.1% Triton X-100) for 5 min. Next, the samples were blocked for 1 hr at room temperature using 5% normal goat serum (Thermo Fisher Scientific). Cells were incubated overnight at 4 °C with primary antibody in blocking solution. The next day, the coverslips were washed three times with PBS. The cells were then incubated with secondary antibody diluted in blocking solution for 1 hour at room temperature. Following this, the cells were washed three times with PBS, stained with DAPI (Thermo Fisher Scientific), and mounted onto slides using Prolong Diamond antifade reagent (Thermo Fisher Scientific).

## Microscopy

All imaging experiments were performed at the Augusta University Cell Imaging Core (RRID:SCR_026799). Fixed images were captured on either an inverted Leica Stellaris confocal microscope or an inverted Zeiss LSM780 equipped with Airyscan. Live imaging of SiR-labeled lysosomes was performed on an inverted Nikon AXR confocal microscope equipped with the NSPARC detector. Images were processed for presentation using Fiji, Adobe Photoshop, and Adobe Illustrator.

## Quantifications

The clustering phenotype of GFP-VPS41 and LAMP1 vesicles (*Figure 3*) was quantified using Imaris 10.2. The nucleus was defined using the 'surface' feature of Imaris and the vesicles were defined using the "spots" feature. The percentage of vesicles that were less than or equal to 10 microns from the nucleus and the percentage of vesicles that were present a distance greater than 10 microns from the nucleus was determined using object to object classification and the filtering feature in Imaris. The quantification of lysosome motility in live cells (*Figure 3—figure supplement 1L*) was also performed in Imaris. Vesicles were defined using the "spots" feature and were tracked over time. Motile vesicles were defined as those that displayed movement over at least a 3 second interval and had at least 0.1 microns of displacement over the course of the imaging experiment. This enabled us to quantify the percent of motile particles per cell and the velocity of these vesicles. The centrosome enrichment phenotype shown in *Figure 4B–E* was quantified by determining the ratio of the mean BICD2 signal in the entire cell vs the mean intensity in the centrosomal area demarcated by Pericentrin. This was done using Imaris. The peroxisome clustering phenotype (*Figure 4H*) was quantified using the "cells" module of Imaris. Within this module, the nucleus was defined using the DAPI channel and the cell boundaries were determined using phalloidin staining of F-actin. Next, the peroxisomes were defined using the "spots" feature built into this module. This enabled us to calculate the average distance of the peroxisomes to the nucleus on a cell-by-cell basis. The cell body enrichment of BICD2 signal (*Figure 5E*) was determined using the "filaments" module in Imaris. This was used to define the cell body and the axon. The mean intensity of signal in the cell body was compared to the mean intensity of signal in the axon. The co-localization between GRAMD1A and BICD2 (*Figure 7*) was determined using the 'co-localization' module in Imaris. Axon length (*Figure 5F*) was quantified using the Simple neurite tracer plugin for FIJI. Densitometry for western blot images captured on the Bio-Rad Chemidoc MP was analyzed using the Bio-Rad Image Lab software. The binding level was compared to the amount of the respective protein detected in the BICD2_wt lane. Graphing of data and statistical analysis was performed using GraphPad Prism 10.

## Acknowledgements

We are grateful to Drs. Erika Holzbaur, Adam Fenton, Juan Bonifacino, Addanki Tirumala, and Raffaella de Pace for advice on transfection and culturing of hippocampal neurons. We are also grateful to Pritha Bagchi for her assistance with the mass spectrometry analysis. This work was supported by grants from the National Institutes of Health to XF (NEI, EY032488) and GBG (NIGMS, R35GM145340). This work was supported in part by the Emory Integrated Proteomics Core (RRID:SCR_023530) and the Augusta University Cell Imaging Core (RRID:SCR_026799).

## Additional information

### Funding

| Funder | Grant reference number | Author |
|--------|------------------------|--------|
| National Institutes of Health | R35GM145340 | Graydon B Gonsalvez |
| National Institutes of Health | EY032488 | Xingjun Fan |

The funders had no role in study design, data collection and interpretation, or the decision to submit the work for publication.

### Author contributions

Hannah Neiswender, Conceptualization, Formal analysis, Investigation, Methodology, Writing – review and editing; Jessica E Pride, Phylicia Allen, Formal analysis, Investigation, Writing – review and editing; Rajalakshmi Veeranan-Karmegam, Data curation, Investigation; Grace Neiswender, Formal analysis, Methodology; Avneesh Prabakar, Formal analysis, Investigation; Caili Hao, Resources, Methodology; Xingjun Fan, Resources, Investigation; Graydon B Gonsalvez, Conceptualization, Resources, Formal analysis, Investigation, Writing – original draft, Project administration, Writing – review and editing

### Author ORCIDs

Phylicia Allen ⓘ https://orcid.org/0000-0002-4040-6246
Graydon B Gonsalvez ⓘ https://orcid.org/0000-0002-4458-8497

Reviewer #1 (Public review): https://doi.org/10.7554/eLife.107503.3.sa1
Reviewer #2 (Public review): https://doi.org/10.7554/eLife.107503.3.sa2
Reviewer #3 (Public review): https://doi.org/10.7554/eLife.107503.3.sa3
Author response https://doi.org/10.7554/eLife.107503.3.sa4

## Additional files

### Supplementary files

MDAR checklist

Supplementary file 1. The wild-type BICD2 interactome.

Supplementary file 2. The interactome of BICD2_N188T vs BICD2_wt.

Supplementary file 3. The interactome of BICD2_R694C vs BICD2_wt.

Supplementary file 4. The interactome of BICD2_R747C vs BICD2_wt.

### Data availability

The raw mass spectrometry data has been deposited to the MASSIVE server and can be accessed using the following ID: MSV000099881.

The following dataset was generated:

| Author(s) | Year | Dataset title | Dataset URL | Database and Identifier |
|-----------|------|---------------|-------------|-------------------------|
| Gonsalvez G | 2025 | Neiswender eLife_2025 protoemics dataset | https://doi.org/10.25345/C57M04D1D | MassIVE, 10.25345/C57M04D1D |

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

# Appendix 1

## Appendix 1—key resources table

| Reagent type (species) or resource | Designation | Source or reference | Identifiers | Additional information |
|---|---|---|---|---|
| Cell line (*Homo sapiens*) | HeLa | ATCC | HeLa (ATCC CCL-2) | |
| Cell line (*Cercopithecus aethiops*) | COS-7 | ATCC | CRL-1651 | |
| Cell line (*H. sapiens*) | Flp-In–293 Cell Line | Thermo Fisher | R75007 | |
| Biological sample (*Rattus norvegicus*) | E18 Rat hippocampus | Transnetyx | KTSDEHP | |
| Antibody | Anti-V5 (mouse monoclonal) | Invitrogen | R960-25 | 1:10,000 for western; 1:1000 for immunofluorescence |
| Antibody | Anti-VPS41 (rabbit polyclonal) | Abcam | ab181078 | 1:1000 for western |
| Antibody | Anti-VPS16 (rabbit polyclonal) | Proteintech | 17776–1-AP | 1:1000 for western |
| Antibody | Anti-VPS18 (rabbit polyclonal) | Proteintech | 67590–1-Ig | 1:5000 for western |
| Antibody | Anti-RANBP2 (mouse monoclonal) | Santa Cruz | sc-74518 | 1:400 for western |
| Antibody | Anti-LAMP1 (rabbit polyclonal) | Cell Signaling Technology | 9091T | 1:600 for immunofluorescence |
| Antibody | Anti-DCTN1 (rabbit polyclonal) | ThermoFisher | PA5-21289 | 1:2000 for western |
| Antibody | Anti-DIC (mouse monoclonal) | Millipore Sigma | D5167 | 1:1000 for western |
| Antibody | Anti-Pericentrin (rabbit polyclonal) | Abcam | ab4448 | 1:200 for immunofluorescence |
| Antibody | Anti-CSPP1 (rabbit polyclonal) | Proteintech | 11931–1-AP | 1:500 for western |
| Antibody | Anti-GRAMD1A (rabbit polyclonal) | Novus Biologicals | NBP2-32148 | 1:1000 for western |
| Antibody | Anti-alpha Tubulin (mouse polyclonal) | Millipore Sigma | T6199 | 1:50,000 for western |
| Antibody | Anti-Dynein heavy chain (mouse monoclonal) | Santa Cruz | sc-514579 | 1:1000 for western |
| Antibody | Anti-KIF5B (rabbit polyclonal) | Proteintech | 21632–1-AP | 1:4000 for western |
| Antibody | Anti-BICD2 (mouse monoclonal) | Thermo Fisher | MA5-23522 | 1:1000 for western |
| Antibody | Anti-Importin beta (rabbit polyclonal) | Cell Signaling Technologies | 60769 | 1:1000 for western |
| Antibody | Anti-GM130 (rabbit polyclonal) | Cell Signaling Technologies | 12480 | 1:1250 for immunofluorescence |
| Antibody | Anti-Cyclin B (rabbit polyclonal) | Cell Signaling Technologies | 4138T | 1:200 for immunofluorescence |
| Antibody | Goat-anti mouse Alexa488 | Thermo Fisher | A-11029 | 1:400 for immunofluorescence |
| Antibody | Goat-anti rabbit Alexa488 | Thermo Fisher | A-11034 | 1:400 for immunofluorescence |
| Antibody | Goat-anti mouse Alexa555 | Thermo Fisher | A-21424 | 1:400 for immunofluorescence |
| Antibody | Goat-anti rabbit Alexa555 | Thermo Fisher | A-21428 | 1:400 for immunofluorescence |
| Antibody | Goat-anti mouse HRP | Thermo Fisher | 31430 | 1:5000 for western |
| Antibody | Goat-anti rabbit HRP | Thermo Fisher | 31460 | 1:5000 for western |
| Recombinant DNA reagent | pCDNA5/FRT/TO | Thermo Fisher | V652020 | |
| Recombinant DNA reagent | mRFP-mTrbo cloned into pCDNA5/FRT/TO | This paper | Available upon request | Materials and methods |

*Appendix 1 Continued on next page*

*Appendix 1 Continued*

| Reagent type (species) or resource | Designation | Source or reference | Identifiers | Additional information |
|---|---|---|---|---|
| Recombinant DNA reagent | BICD2_wt-mTrbo cloned into pCDNA5/FRT/TO | This paper | Available upon request | Materials and methods |
| Recombinant DNA reagent | BICD2_delCC3-mTrbo cloned into pCDNA5/FRT/TO | This paper | Available upon request | Materials and methods |
| Recombinant DNA reagent | BICD2_delCC1-mTrbo cloned into pCDNA5/FRT/TO | This paper | Available upon request | Materials and methods |
| Recombinant DNA reagent | BICD2_CC3-mTrbo cloned into pCDNA5/FRT/TO | This paper | Available upon request | Materials and methods |
| Recombinant DNA reagent | BICD2_N188T-mTrbo cloned into pCDNA5/FRT/TO | This paper | Available upon request | Materials and methods |
| Recombinant DNA reagent | BICD2_R694C-mTrbo cloned into pCDNA5/FRT/TO | This paper | Available upon request | Materials and methods |
| Recombinant DNA reagent | BICD2_R747C-mTrbo cloned into pCDNA5/FRT/TO | This paper | Available upon request | Materials and methods |
| Recombinant DNA reagent | GFP-VPS41 | This paper | Available upon request | Materials and methods |
| Recombinant DNA reagent | pEGFP-C3 | Clontech | Discontinued by manufacturer | |
| Recombinant DNA reagent | pLVX-Puro | Takara | 631849 | |
| Recombinant DNA reagent | BICD2_wt-mNeon in pLVX-Puro | This paper | Available upon request | Materials and methods |
| Recombinant DNA reagent | BICD2_N188T-mNeon in pLVX-Puro | This paper | Available upon request | Materials and methods |
| Recombinant DNA reagent | BICD2_R694C-mNeon in pLVX-Puro | This paper | Available upon request | Materials and methods |
| Recombinant DNA reagent | BICD2_R747C-mNeon in pLVX-Puro | This paper | Available upon request | Materials and methods |
| Recombinant DNA reagent | PEX3-mGreenLantern-FKBP in pLVX-Puro | This paper | Available upon request | Materials and methods |
| Recombinant DNA reagent | BICD2_wt-FRB cloned into pCDNA5/FRT/TO | This paper | Available upon request | Materials and methods |
| Recombinant DNA reagent | BICD2_N188T-FRB cloned into pCDNA5/FRT/TO | This paper | Available upon request | Materials and methods |
| Recombinant DNA reagent | BICD2_R694C-FRB cloned into pCDNA5/FRT/TO | This paper | Available upon request | Materials and methods |
| Recombinant DNA reagent | BICD2_R747C-FRB cloned into pCDNA5/FRT/TO | This paper | Available upon request | Materials and methods |
| Recombinant DNA reagent | EGFP-RAB6AQ72L | Addgene | Plasmid #49483 | |
| Recombinant DNA reagent | ER-mScarletI | Addgene | Plasmid #137805 | |
| Recombinant DNA reagent | GFP-BICD2 | Addgene | Plasmid #164626 | |
| Sequence-based reagent | control siRNA | Thermo Fisher | 4390843 | |
| Sequence-based reagent | DYNC1H1 siRNA | Thermo Fisher | 4390824 | Assay ID# s4200 |
| Sequence-based reagent | BICD2 siRNA-1 | Thermo Fisher | 4392420 | Assay ID# s23497 |
| Sequence-based reagent | BICD2 siRNA-2 | Thermo Fisher | 4392420 | Assay ID# s225943 |
| Sequence-based reagent | KIF5B siRNA | Thermo Fisher | 4390824 | Assay ID# s732 |
| Sequence-based reagent | RANBP2 siRNA | Thermo Fisher | 4390824 | Assay ID# s11775 |
| Sequence-based reagent | VPS41 siRNA | Thermo Fisher | 4392420 | Assay ID# s25770 |
| Commercial assay or kit | Streptavidin magnetic beads | Thermo Fisher | 88816 | |

*Appendix 1 Continued on next page*

*Appendix 1 Continued*

| Reagent type (species) or resource | Designation | Source or reference | Identifiers | Additional information |
|---|---|---|---|---|
| Commercial assay or kit | Streptavidin agarose beads | Thermo Fisher | 20357 | |
| Commercial assay or kit | V5 trap beads | Chromotek | v5ta-20 | |
| Commercial assay or kit | SiR lysosome kit | Cytoskeleton, Inc. | CY-SC012 | |
| Chemical compound, drug | D-biotin | Thermo Fisher | B 1595 | |
| Software, algorithm | Imaris 10.1 | Oxford Instruments | RRID:SCR_007370 | |
| Software, algorithm | Prism 10 | GraphPad | RRID:SCR_002798 | Version 10.6.1 |
| Software, algorithm | Fiji | ImageJ 1.54 p | RRID:SCR_002285 | |

