## [Editor Report · eLife Assessment]

In their study, Neiswender et al. provide **important** insights into how BicD2 variants linked to spinal muscular atrophy alter dynein activity and cargo specificity. The authors present **convincing** evidence that disease-associated mutations lead to interactome changes, supported by additional validation of the BicD2/HOPS complex and discussion of their functional implications. This well-executed study offers invaluable datasets and a strong foundation for future exploration of disease mechanisms.

---

## [Referee Report · Reviewer #1 (Public review)]

In this work, Neiswender and colleagues test the hypothesis that mutations in BicD2 that are associated with SMALED alter BicD2-cargo interactions. To do this, they first establish the WT BicD2 cargo interactome (using a proximity-dependent biotin ligase screen with Turbo-ID on the BicD2 C-terminus). In addition to known cargo interactors, they also identified many proteins in the HOPs complex. Interestingly, they find that the HOPs complex may interact with BicD2 in a different manner than other known cargos. The authors also show that while BicD2 is required for the HOPs complex localization, on average, depletion of BicD2 from HeLa and Cos7 cells causes HOPs and Lysosome mislocalization that is consistent with Kinesin-1 trafficking defects, rather than dynein. The authors also use proximity biotin ligase approaches to define the cargo interactome of three BicD2 variants associated with SMALED. One variant (R747C) has the most altered cargo interactome. The authors highlight one protein, in particular, GRAMD1A, that is only found in the R747C dataset and mislocalizes specifically when R747C is expressed.

The work in this manuscript is of a very high quality and contributes important findings to the field.

Comments on revisions:

The authors did a great job addressing the points I brought up!

---

## [Referee Report · Reviewer #2 (Public review)]

Neiswender et al. investigated the interactomes between wild-type BICD2 and BICD2 mutants that are associated with Spinal Muscular Atrophy with Lower Extremity Predominance (SMALED2). Although BICD2 has previously been implicated in SMALED2, it is unclear how mutations in BICD2 may contribute to disease symptoms. In this study, the authors characterize the interactome of wild-type BICD2 and identify potential new cargos including the HOPS complex. The authors then chose three SMALED2-associated BICD2 mutants and compared each mutant interactome to that of wild-type BICD2. Each mutant had a change in the interactome, with the most drastic being BICD2_R747C, a mutation in the cargo binding domain of BICD2. This mutant displayed less interaction with a potential new BICD2 cargo, the HOPS complex. Additionally, it displayed more interaction with an ER protein, GRAMD1A.

The data in the paper is generally strong but the major conclusions of this paper need more evidence to be better supported.

(1) The authors use cells that have been engineered to express the different BICD2 constructs. As shown in Figure 4B, the authors see wide expression of BICD2_WT throughout the cell. However, WT BICD2 usually localizes to the TGN. This widespread localization introduces some uncertainty about the interactome data. The authors should either try to verify the interaction data (specifically with the HOPS complex and GRAMD1A) by immunoprecipitating endogenous BICD2 or by repeating their interactome experiment in Figure 1 using BICD2 knockout cells that express the BICD2_WT construct. This should also be done to verify the immunoprecipitation and microscopy data shown in Figure 7.

(2) The authors conclude that cargo transport defects resulting from BICD2 mutations may contribute to SMALED2 symptoms. However, the authors are unable to determine if BICD2 directly binds to the potential new cargo, the HOPS complex. To address this, the authors could purify full-length WT BICD2 and perform in vitro experiments. Furthermore, the authors were unable to identify the minimal region of BICD2 needed for HOPS interaction. The authors could expand on the experiment attempted with the extended BICD2 C-terminal using a deltaCC1 construct, which could also be used for in vitro experiments.

(3) Again, the authors conclude that BICD2 mutants cause cargo transport defects that are likely to lead to SMALED2 symptoms. This would be better supported if the authors are able to find a protein relevant to SMALED2 and examine if/how its localization is changed under expression of the BICD2 mutants. The authors currently use the HOPS complex and GRAMD1A as indicators of cargo transport defects, but it is unclear if these are relevant to SMALED2 symptoms.

Comments on revisions:

The investigators did a good job in responding to our initial concerns (see below). We appreciate that they used siRNA to address our first comment because they do not have a BICD2 KO cell line. We appreciated that they added a new section in the Discussion to address the limitations of the study.

In regards to our first comment about the BICD2 WT construct localization, since they use KD to validate the interaction between their BICD2 WT construct and VPS41, it would be nice to see localization of this construct under the KD condition. However, the binding they presented in Sup. Fig 1B does look convincing, so this may not be necessary.

Overall, I believe this revision has satisfied our previous concerns.

---

## [Referee Report · Reviewer #3 (Public review)]

Summary:

BicD2 is a motor adapter protein that facilitates cellular transport pathways, which are impacted by human disease mutations of BicD2 causing spinal muscular atrophy with lower extremity dominance (SMALED2). The authors provide evidence that some of these mutations result in interactome changes, which may be the underlying cause of the disease. This is supported by proximity biotin ligation screens, immunoprecipitation and cell biology assays. The authors identify several novel BicD2 interactions such as the HOPS complex that participates in the fusion of late endosomes and autophagosomes with lysosomes, which could have important functions. Three BicD2 disease mutants studied had changes in the interactome, which could be an underlying cause for SMALED2. The study extends our understanding of the BicD2 interactome under physiological conditions, as well as of the changes of cellular transport pathways that result in SMALED2. It will be of great interest for the BicD2 and dynein fields.

Strengths:

Extensive interactomes are presented for both WT BicD2 as well as the disease mutants, which will be valuable for the community. The HOPS complex was identified as a novel interactor of BicD2, which is important for fusion of late endosomes and lysosomes, which is of interest, since some of the BicD2 disease mutations result in Golgi-fragmentation phenotypes. The interaction with the HOPS complex is affected by the R747C mutation, which also results in a gain of function interaction with GRAMD1A.

Weaknesses:

The manuscript should be strengthened by further evidence of the BicD2/HOPS complex interaction and the functional implications for spinal muscular atrophy by changes in the interactome through mutations. Which functional implications does the loss of the BicD2/HOPS complex interaction and the gain of function interaction with GRAMD1A have in the context of the R747C mutant?

Major points:

(1) In the biotin proximity ligation assay, a large number of targets were identified, but it is not clear why only the HOPS complex was chosen for further verification. Immunoprecipitation was used for target verification, but due to the very high number of targets identified in the screen, and the fact that the HOPS complex is a membrane protein that could potentially be immunoprecipitated along with lysosomes or dynein, additional experiments to verify the interaction of BicD2 with the HOPS complex (reconstitution of a complex in vitro, GST-pull down of a complex from cell extracts or other approaches) are needed to strengthen the manuscript.

(2) In the biotin proximity ligation assay, a large number of BicD2 interactions were identified that are distinct between the mutant and the WT, but it was not clear why particularly GRAMD1A was chosen as gain of function interaction, and what the functional role of a BicD2/GRAMD1A interaction may be. A Western blot shows a strengthened interaction with the R747C mutant but GRAMD1A also interacts with WT BicD2.

(3) Furthermore, functional implications of changed interactions with HOPS and GRAMD1A in the R747C mutant are unclear. Additional experiments are needed to establish the functional implication of the loss of the BicD2/HOPS interaction in the BicD2/R747C mutant. For the GRAMD1A gain of function interaction, according to the authors a significant amount of the protein localized with BicD2/R747C at the centrosomal region. This changed localization is not very clear from the presented images (no centrosomal or other markers were used, and the changed localization could also be an effect of dynein hyper activation in the mutant). Furthermore, the functional implication of a changed localization of GRAMD1A is unclear from the presented data.

Comments on revisions:

After a major revision, the manuscript is much improved. Additional evidence for the HOPS complex/BicD2 interaction was provided (the interaction was identified in multiple independent screens), and while the authors unfortunately were not able to confirm a direct interaction between BicD2 and the HOPS complex, additional caveats were added in the result section, which clearly state these limitations. The authors also included a very nice discussion of potential physiological roles of the GRAMD1A mislocalization in the disease mutant, which could potentially affect cholesterol transport and homostatis. Limitations of the presented approaches were clearly described as caveats.

---

## [Author Response]

The following is the authors’ response to the original reviews.

**Reviewer #1 (Public review):**

(1) I was surprised at the effect of BicD2 knockdown on LAMP (and VPS41) localization, which really suggests that in HeLa and Cos7 cells, BicD2 regulation of Kinesin-1 (rather than dynein) is the primary driver of lysosome localization. The KIF5B-knockout rescue of the BicD2overexpression phenotype was a very powerful result that supports this conclusion. Have the authors looked at other cargos, eg, Golgi or centrosomes in G2? Can the authors include more discussion about what this result means or how they imagine dynein and kinesin-1's interaction with BicD2 is regulated?

We have performed this experiment as requested by the reviewer. The BICD2 siRNA also resulted in Golgi fragmentation and localization defects of the centrosome in cells that are in G2 phase of the cell cycle (Supplemental Fig. 2E-H).

We have also added additional discussion related to how BICD2 might couple cargos to opposite polarity motors (lines 440-447). Interestingly, the lysosome motility defect we observe upon BICD2 knock down has similarity to the RAB6A trafficking phenotype. In both cases, what one sees is a sharp reduction in the number of motile particles rather than a reversal in the direction of motility. This suggests that both motors are involved in the steady state distribution of these cargoes.

(2) Have the authors examined if the SMALED mutants show diminished or increased binding to KIF5B? While the authors are correct that the mutations could hyperactivate dynein because they reduce BicD2 autoinhibition, it is possible that the SMALED mutants hyperactivate dynein because they no longer bind kinesin. This would be particularly interesting, given the complex relationship between BicD2 regulation of dynein and kinesin that the authors show in Figure 3.

Thank you for this suggestion. We had not considered this. We have added this experiment in the revised manuscript (Supplemental Fig. 3H, I). We find that the interaction between wild-type BICD2 and KIF5B is only slightly above the control. This is consistent with published findings that indicate that although the isolated CC2 domain of BICD2 is able to interact with KIF5B, the binding is lower for the full-length protein. This is most likely due to the intramolecular interaction between the N and C-termini of BICD2 partially blocking the binding site. Interestingly, however, all three mutants display a reduced interaction with KIF5B, with the reduction being most severe for the cargo domain binding mutants. Thus, as we discuss in the revised manuscript, dynein hyperactivity likely results from increased binding to dynein and a concurrent reduction in binding to KIF5B.

(3) What is already known about the protein GRAMD1A? Did the authors choose to focus on GRAMD1A because it was the only novel interaction found in the SMALED mutant interactomes, or was this protein interesting for a different reason? Does the known function of GRAMD1A explain the potential dysfunction of cells expressing BICD2_R747C or patients who have this mutation? More discussion of this protein and why the authors focused on it would really strengthen the manuscript.

We chose to focus on GRAMD1A for a few reasons. The protein that displayed the highest gain of function interaction with BICD2_R747C in our proteomic analysis was Plastin. However, using at least one antibody against Plastin, we were not able to validate this result. In addition, we had previously performed a proteomic screen using a BICD2_R747A (arginine to alanine) mutation and had compared the interactome of this mutant to the wild-type protein. Plastin was not recovered in that screen but the top hit was GRAMD1A. Given that we isolated GRAMD1A in two separate screens as a gain of function interaction, we believed the result was worth focusing on for followup studies.

GRAMD1A (as well as its paralogs GRAMD1B and C) function in non-vesicle transport of accessible cholesterol from the plasma membrane to the ER. We have added additional discussion on GRAMD1A (lines 484-495). While we observe a relocalization of GRAMD1A in mutant expressing cells, we do not know whether this is sufficient to result in cholesterol transport defects. There are several routes for cholesterol uptake, with the GRAMD1A pathway representing just one these routes.

**Reviewer #2 (Public review):**
(1) The authors use cells that have been engineered to express the different BICD2 constructs. As shown in Figure 4B, the authors see wide expression of BICD2_WT throughout the cell. However, WT BICD2 usually localizes to the TGN. This widespread localization introduces some uncertainty about the interactome data. The authors should either try to verify the interaction data (specifically with the HOPS complex and GRAMD1A) by immunoprecipitating endogenous BICD2 or by repeating their interactome experiment in Figure 1 using BICD2 knockout cells that express the BICD2_WT construct. This should also be done to verify the immunoprecipitation and microscopy data shown in Figure 7.

The localization of our exogenous BICD2-mNeon constructs is similar to what others have seen using GFP tagged versions of the protein (for example Peeters et al., 2013). In addition, in the experiment shown in the initial version of the paper, we were focusing on the centrosomal localization of BICD2. However, our BICD2-mNeon construct is also observed at the Golgi, in addition to its localization throughout the cell (Supplemental Fig. 3C).

We attempted to perform a co-immunoprecipitation experiment using endogenous proteins as suggested by the reviewer. Although a rabbit polyclonal antibody was able to coimmunoprecipitate RANBP2 with BICD2, the antibody complex of heavy and light chains comigrated with the VPS41 band and was abundantly detected by the secondary antibody used in the western blot. Thus, we were not able to make a conclusion regarding whether or not VPS41 was present in the co-immunoprecipitate. We attempted the experiment using a mouse monoclonal antibody against BICD2. However, this antibody failed in the immunoprecipitation experiment and we could not detect either RANBP2 (a validated cargo) or VPS41. Although the VPS41 antibody we used in the paper works for western blot, it does not recognize the native protein. Thus, despite our best efforts, we are not able to draw a valid conclusion from these coip experiments.

It is beyond the scope of the revision to perform the entire experiment in a BICD2 KO cell line. A BICD2 KO cell line does not exist and it would take several months to make such a knock out in the FLP IN HEK cells that were used in this manuscript. However, we have validated the interaction between BICD2 and VPS41 in cells that have been depleted of endogenous BICD2 (Supplemental Fig. 1B). The transgenic constructs contain silent mutations that make them refractory to bicD2 siRNA1. Thus, although endogenous BICD2 is depleted by the siRNA treatment, wild-type and mutant BICD2_TurboID is not. A similar approach was also used to demonstrate the gain of function interaction between BICD2_R747C and GRAMD1A in cells depleted of endogenous BICD2 (Supplemental Fig. 5A).

(2) The authors conclude that cargo transport defects resulting from BICD2 mutations may contribute to SMALED2 symptoms. However, the authors are unable to determine if BICD2 directly binds to the potential new cargo, the HOPS complex. To address this, the authors could purify full-length WT BICD2 and perform in vitro experiments. Furthermore, the authors were unable to identify the minimal region of BICD2 needed for HOPS interaction. The authors could expand on the experiment attempted with the extended BICD2 C-terminal using a deltaCC1 construct, which could also be used for in vitro experiments.

We have not been successful in purifying full length BICD2 in bacteria, perhaps due to solubility issues. However, we have added several experiments to further examine the nature of the BICD2-HOPS complex interaction.

We have performed the experiment as requested. We find that BICD2_delCC1 is able to bind VPS41, but not as efficiently as the full length protein. However, unlike the CC3 cargo binding construct, the BICD2_delCC1 construct also displays reduced binding to RANBP2 (Supplemental Fig. 1D). We attribute this defect to either the intramolecular BICD2 interaction blocking cargo binding or potentially to a folding defect in the BICD2_delCC1 construct. Thus, although we performed this experiment as suggested by the reviewer, we are not able to make a solid conclusion.

Based on the fact that VPS41 was the most abundantly detected HOPS component in the BICD2 interactome, we hypothesized that it was the point of direct contact between BICD2 and the HOPS complex. However, contrary to our hypothesis, depletion of VPS41 did not compromise the association between BICD2 and VPS16 and VPS18 (Supplemental Fig. 1E). Thus, we conclude that there are multiple points of contact between BICD2 and the HOPS complex, with BICD2 perhaps recognizing a common motif or domain present in these proteins.

We next attempted to map the interaction site using Alphafold2 multimer. Although we were able to use this platform to predict a high confidence interaction between BICD2 and RAB6A (consistent with published results), this did not yield a high confidence prediction for the BICD2HOPS complex interaction.

Ultimately although we added several new experiments, we were not able to determine the minimal region for binding, nor whether the interaction is direct or indirect. These caveats are clearly stated in the revised manuscript. Regardless of whether the interaction is direct or indirect however, it is noteworthy that the association between BICD2 and the HOPS complex is reduced by the R747C SMALED2 mutation.

(3) Again, the authors conclude that BICD2 mutants cause cargo transport defects that are likely to lead to SMALED2 symptoms. This would be better supported if the authors are able to find a protein relevant to SMALED2 and examine if/how its localization is changed under expression of the BICD2 mutants. The authors currently use the HOPS complex and GRAMD1A as indicators of cargo transport defects, but it is unclear if these are relevant to SMALED2 symptoms.

This point was addressed in the general discussion. Given the complexity of SMALED2 (autosomal dominant disorder; variable phenotypic severity; adult onset disorder in many instances, etc.) it is very hard to model in a cell line. One of the reasons we focused our studies on the HOPS complex and VPS41 in particular was because mutations in VPS41 are associated with spinocerebellar ataxia, a neurodevelopment disorder. However, we cannot conclude whether the reduction/loss of interaction of BICD2 with the HOPS complex is causative for disease symptoms. We also cannot conclude at present whether the mis-targeting of GRAMD1A is causative for disease symptoms. We have discussed these caveats in the revised manuscript and have included a section in the discussion that specifically lists the limitations of our study (lines 511-530).

With that said, we can conclude that mutations in the cargo binding domain of BICD2 result in dynein hyperactivity, altered BICD2 localization in hippocampal neurons, and reduced neurite growth. Given that we observe interactome changes in HEK cells, it is plausible that interactome changes also exist in motor neurons. However, even in the absence of interactome changes, hyperactivation of dynein alone can result in cargo trafficking defects; the same cargos can be excessively localized in the soma vs the axon. As noted previously, however, a thorough examination of these points will require the use of genetically engineered motor neurons and is beyond the scope of the current study.

**Reviewer #3 (Public review):**
Strengths:Extensive interactomes are presented for both WT BicD2 as well as the disease mutants, which will be valuable for the community. The HOPS complex was identified as a novel interactor of BicD2, which is important for fusion of late endosomes and lysosomes, which is of interest, since some of the BicD2 disease mutations result in Golgi-fragmentation phenotypes. The interaction with the HOPS complex is affected by the R747C mutation, which also results in a gain-of-function interaction with GRAMD1A.Weaknesses:The manuscript should be strengthened by further evidence of the BicD2/HOPS complex interaction and the functional implications for spinal muscular atrophy by changes in the interactome through mutations. Which functional implications does the loss of the BicD2/HOPS complex interaction and the gain of function interaction with GRAMD1A have in the context of the R747C mutant?(1) In the biotin proximity ligation assay, a large number of targets were identified, but it is not clear why only the HOPS complex was chosen for further verification. Immunoprecipitation was used for target verification, but due to the very high number of targets identified in the screen, and the fact that the HOPS complex is a membrane protein that could potentially be immunoprecipitated along with lysosomes or dynein, additional experiments to verify the interaction of BicD2 with the HOPS complex (reconstitution of a complex in vitro, GST-pull down of a complex from cell extracts or other approaches) are needed to strengthen the manuscript.

As discussed for reviewer 2 (point 2), we have added several experiments to better characterize the BICD2-HOPS complex interaction.

We chose to focus on the HOPS complex for a few reasons. The list of interactions that displayed a >2 fold enrichment vs control was actually not that large (66 proteins). Within this list, we identified 4 out of 6 HOPS components and VPS41 was the 5th most enriched protein in the BICD2 interactome (RANBP2 by contrast was #16 on this list). Furthermore, the BICD2_R747C mutation resulted in greatly reduced interaction of BICD2 with the HOPS complex, whereas its interaction with dynein was increased. These results indicate that these proteins are not simply immunoprecipitating with the BICD2/dynein complex. Apart from the HOPS complex, lysosomal proteins were not present in the interactome, making it unlikely that they were identified due to non-specific interactions between BICD2 and co-precipitating lysosomes.

(2) In the biotin proximity ligation assay, a large number of Bi cD2 interactions were identified that are distinct between the mutant and the WT, but it was not clear why, particularly GRAMD1A was chosen as a gain-of-function interaction, and what the functional role of a BicD2/GRAMD1A interaction may be. A Western blot shows a strengthened interaction with the R747C mutant, but GRAMD1A also interacts with WT BicD2.

Please see the above discussion on GRAMD1A (reviewer 1, point 3). GRAMD1A comes down non-specifically with the binding control as well as BICD2_wt. We therefore conclude that wildtype BICD2 does not specifically interact with GRAMD1A above background levels (Fig. 7, compare the control lane vs BICD2-wt).

(3) Furthermore, the functional implications of changed interactions with HOPS and GRAMD1A in the R747C mutant are unclear. Additional experiments are needed to establish the functional implication of the loss of the BicD2/HOPS interaction in the BicD2/R747C mutant. For the GRAMD1A gain of function interaction, according to the authors, a significant amount of the protein localized with BicD2/R747C at the centrosomal region. This changed localization is not very clear from the presented images (no centrosomal or other markers were used, and the changed localization could also be an effect of dynein hyperactivation in the mutant). Furthermore, the functional implication of a changed localization of GRAMD1A is unclear from the presented data.

We have performed the experiment as requested by the reviewer. The re-localized GRAMD1A localizes adjacent to Pericentrin, a centrosomal marker (Supplemental Fig. 5B-F). GRAMD1A and BICD2 appear to co-localize in a ring around the Pericentrin marked centrosome.

The re-localization of GRAMD1A to the centrosomal area by BICD2_R747C appears to be unique to this mutant, and not simply an issue of dynein hyperactivity. The other two mutants tested, BICD2_N188T and BICD2_R694C also hyperactivate dynein. However, they do not result in the same type of dramatic re-localization of GRAMD1A as we observe with the BICD2_R747C mutant. We conclude that this altered localization results from a gain of function interaction with BICD2_R747C as well as dynein hyperactivity.

**Reviewer #1 (Recommendations for the authors):**
Please add a discussion about how the authors calculated the Cell Body enrichment shown in 5E. Is this a ratio of the BicD2 intensity in the cell body:axon? Did the authors normalize for potential differences in BicD2 variant expression?

Yes, it is a ratio of the intensity between the cell body and axon. This is described in the Methods section under quantification (lines 725-728). We attempted to image cells expressing similar amounts of protein.

**Reviewer #2 (Recommendations for the authors):**
(1) The paper would benefit from an explanation of why the authors chose to follow up on the HOPS complex out of all proteins identified in the interactome experiment.

This discussion has been included in the revised manuscript.

(2) In panel B of Supplementary Figure 1, RFP mTurbo has a significant amount of non-specific binding to VPS18. The authors note that in the initial interactome experiment, there was a twofold enrichment of this protein in BICD2 pulldown versus control. Do the authors have a co-IP that has a similar enrichment?

VPS18 occasionally comes down non-specifically with our RFP-TurboID control. However, the interaction is specific, because very little VPS18 comes down with the BICD2 construct lacking the cargo binding domain (Fig. 2B). An additional example of the VPS18 binding result is shown in Supplemental Fig. 1E.

(3) In Figure 2B, there seems to be less Vps18 in the input for BICD2 delCC3-mTrbo. Do the authors have a blot where there is equal input across all conditions? This may increase the slight signal seen in the pulldown.

The blot shown in Supplemental Fig. 1C has equivalent load for VPS18 across all lanes. Minimal binding of VPS18 is observed with the BICD2_delCC3 sample.

(4) In Figure 3, can the authors show representative images of GFP-VPS-41 and LAMP1 localization that are at the same magnification? It currently looks as if the localization pattern differs between the two under control siRNA. Alternatively, the authors should show colocalization of the two, as the authors note both are localized to late endosomes/lysosomes.

We have provided additional images that are at the same magnification (Supplemental Fig. 2IK). Co-localization between GFP-VPS41 (rabbit polyclonal antibody against GFP) and LAMP1 (rabbit polyclonal antibody) is not possible. However, published studies have shown that a subset of V5 tagged VPS41 vesicles are positive for LAMP1. We have cited this study.

(5) In Supplementary Figure 2, the authors should show the knockdown efficiency of both BICD2 siRNAs. The VPS41 staining in panel B looks like there is less perinuclear localization than with BICD2 siRNA 1. Is the because of knockdown efficiency?

We have included this data (Supplemental Fig. 2B). Both siRNAs are capable of depleting BICD2. However, we do see slightly more effective knock down with siRNA-1.

(6) The data in Figure 4A would be more striking with quantification.

Quantifications have been provided (Supplemental Fig. 3A,B). Using a one-way Anova analysis, BICD2_R747C is the only mutant that shows significance. Variability in the binding experiment resulted in the other two mutants not showing a statistically significant change. However, the additional assays that are provided (centrosomal enrichment of BICD2 and peroxisome tethering) clearly demonstrate that the R694C mutant also results in dynein hyperactivation. It should be noted that the analysis done by Huynh et al., 2017 also showed a binding increase between BICD2 disease mutants and dynein. However, due to binding variability, their results were not not statistically significant.

(7) Can the authors explain how centrosome enrichment is calculated in Figure 4F? The intensity of colocalization with the centrosome between mutant constructs visually does not look significantly different. Is this a ratio of centrosome localization to cell body localization?

We apologize for this omission. This has been added to the quantification section of the Methods (lines 721-723). Yes, it is a ratio of mean signal at the centrosome vs mean signal in the rest of the cell.

(8) The current input blot in Supplementary Figure 4A shows increasing amounts of importin beta across the lanes. Do the authors have a blot of panel A in which the input level of importin beta is the same between constructs? Does this change the level of importin beta that is pulled down?

Another replicate of this experiment has been shown. We have retained the original experiment as well (Supplemental Figs. 4A, B).

**Reviewer #3 (Recommendations for the authors):**
Minor points:(1) In the .pdf version of the supplemental tables, the text is often cropped. It is recommended to delete the .pdf versions and just retain the Excel versions of the tables.

We are not sure why this occurred. Excel files were provided. In addition, the raw data from the mass spectrometry experiments will also be included with the final version of the manuscript.

(2) Line 367: For transport of Rab6, kinesin-1 is the dominant motor, but dynein is still active and engaging in a tug of war (Serra Marquez et al 2022).

Thank you. We have revised our text to include this discussion. In this regard, LAMP1 vesicles are similar. Loss of BICD2 results in a greater number of stationary vesicles rather than vesicles that are excessively targeted towards the microtubules minus end.

(3) Line 371: BicD2 is required for the transport of RanBP2 from annulate lamellae to nuclear pore complexes.

Thank you. We have modified our text.

(4) Yi et al., 2023 have previously shown changed interactions of the BicD2/R747C mutant, such as decreased binding to Nup358 and increased binding to Nesprin-2, as well as functional implications for the associated brain developmental pathways, which should be acknowledged.

We apologize for leaving this out. In the original version of the manuscript, we were attempting to keep the discussion more concise. We have added a discussion of these findings in the revised manuscript (lines 496-507).